# MODEL GROWTH SCHEDULE LEARNING VIA OPTIMAL PATH (SLOP) FOR EFFICIENT LLM PRE-TRAINING

## ABSTRACT

Existing training methods for Transformer-based large language models (LLMs) rely on massive amounts of data training from scratch, which requires a high cost in terms of compute and time. Recent studies have demonstrated the great potential of improving the LLM's training efficiency by growing from small pre-trained models to large ones—a technique known as model growth. There are two main research problems associated with model growth: growth schedule and growth operators. Existing research focuses on growth operators, detailing specific manipulations of potential dimensions to expand Transformer parameters. Few studies have investigated the optimal growth schedule, which involves integrating all possible growth operators to create an optimal multi-staged growth path. This work introduces SLOP, a **S**chedule **L**earning methodology via an **O**ptimal **P**ath for multi-stage growth of models with minimal experimental training. SLOP utilizes marginal utility as an appropriate measure for an optimal schedule that balances training costs and model performance after multi-stage growth. With this measurement, the objective of determining the optimal growth schedule is converted into a dynamic programming problem, which is then solved mathematically in polynomial time. Experimental results illustrate SLOP's theoretical validity as well as its efficiency, outperforming alternative schedules in a range of settings.

## 1 INTRODUCTION

Transformer-based large language models (LLMs), such as GPT (Radford et al., 2019) and T5 (Raffel et al., 2020) have demonstrated impressive emergent abilities across various tasks. Existing LLM training methods, on the other hand, require enormous amounts of data training from scratch, which is both computationally and time costly. To save costs, there is a growing interest in effective pre-training paradigms. One promising research direction (Chen et al., 2021; Wang et al., 2023; Yao et al., 2023) is growing from small pre-trained models to large ones, known as model growth. In practice, Li et al. (2023) trained a large-scale 100B-parameter model from a 16B-parameter model utilizing model growth methods.

There are two main research problems associated with model growth: determining the optimal growth schedule, and designing efficient growth operators (Yao et al., 2023). Existing works (Gu et al., 2021; Chen et al., 2021; Wang et al., 2023; Chen et al., 2015) primarily focus on growth operators, detailing specific manipulations of potential dimensions (such as layers, hidden states, etc.) to expand Transformer parameters. They also investigate ways to inherit knowledge from the smaller model by developing initialization methods for the newly extended parameters and taking the entire training state as input (e.g., the optimizer state, the learning rate schedule).

Research on the growth schedule is limited. The methodology involves integrating all possible growth operators to create an optimal multi-staged growth path. At each stage, one dimension is expanded to develop an intermediate structure until the entire target LLM structure is attained. Existing works either adopt a single-stage growth without consideration for model schedules (Gong et al., 2019; Gu et al., 2021), or focus on determining the optimal scheduling using empirical insights, even though it may be theoretically suboptimal (Shen et al., 2022; Yao et al., 2023). Ultimately, establishing an optimal schedule for multi-staged model growth necessitates consideration of several fundamental challenges. **1)** What is an appropriate measurement for an optimal schedule? **2)** How are growth operators implemented sequentially, and what number of parameters are inserted at each

stage? **3)** The exponential search space required for trial training makes it prohibitively expensive to explore all possible growth paths (from the initial small model to the target large model) in order to choose the most optimal one.

To address the aforementioned issues, we present a **S**chedule **L**earning methodology via **O**ptimal **P**ath, abbreviated as SLOP, for multi-stage growth of models with limited experimental training efforts. For Transformer-based LLMs, SLOP considers all possible expansion dimensions, expanding one dimension per stage, but ignores more complex cases where several dimensions compound to increase per stage, leading to a search space burst. It is worth noting that within our framework, depth-only or width-only growth may be considered a specific case.

Specifically, we formulate the problem of finding the optimal schedule for multi-stage model growth. The marginal utility (Samuelson, 1937) is used as an appropriate measure for an optimal schedule that balances training costs and model performance after multi-stage growth. With this measurement, we can consider the task of determining an optimal growth schedule as a dynamic programming problem. Finally, we demonstrate that the dynamic programming problem enables the theoretical resolution of an optimal schedule in polynomial time, reducing the computational effort required for trial training within the exponential search space.

To validate the correctness of SLOP's theoretical reasoning results, we conduct experiments by expanding various starting model sizes (e.g., 100M, 450M) to 1 billion decoder-only target LLMs. It also shows that SLOP outperforms alternative schedules in a variety of scenarios, resulting in a reduction in computational usage. Further ablation studies are conducted to evaluate our approach on various growth scenarios.

## 2 RELATED WORK

**Efficient LLM training.** Efficient pretraining of large language models aims to reduce FLOPs. Recent research focuses on stagewise efficient pretraining (Panigrahi et al., 2024), progressive pretraining, or model reusing (Chen et al., 2015; 2021; Wang et al., 2023; Yao et al., 2023). Specifically, model reusing involves maintaining the function of a pre-trained model as its size increases, resulting in an initial state that performs well even when scaled to a larger model. Net2Net (Chen et al., 2015) is the first to introduce the concept of function-preserving transformations in model reusing, expanding the width dimension by splitting neurons and increasing the depth by adding identity layers. Bert2BERT (Chen et al., 2021) applies function-preservation to the Transformer structure, extending Net2Net's concept. LiGO (Wang et al., 2023) recently utilizes a trainable linear operator to develop an efficient expansion strategy. ELLE (Qin et al., 2022) employs function-preserving model expansion within specific domains, leveraging pre-trained domain prompts to efficiently adapt to emerging data over time. Our method concentrates on the model growth schedule, an area that previous studies have rarely addressed.

**Model growth schedule.** The formulation of a model growth schedule is an essential research topic due to the rising prominence of model reuse and progressed pre-training. Gong et al. (2019) and Gu et al. (2021) utilize heuristics rules that divide the training steps into distinct expansion stages to determine the training schedule. Shen et al. (2022) identifies optimal growth schedules that maximize compute savings by applying scaling laws to initiate a new stage when training efficiency decreases. The most relevant work for us is MSG (Yao et al., 2023), which provides empirical insights for constructing an efficient schedule considering all possible growth dimensions. However, they implement an empirical optimal solution that demonstrates practical efficiency, despite the fact it may be theoretically suboptimal. Existing works have not systematically or theoretically explored methodologies for identifying optimal growth schedules by relying instead on empirical results, allowing more space for our method's innovation in the field of optimal growth schedule learning.

## 3 METHODOLOGY

### 3.1 PRELIMINARY

We start by defining some key terms. Consider a model $y = M(x, \theta)$ that takes input $x$ and outputs $y$ with parameters $\theta$. $M$ is the Transformer (Vaswani et al., 2017) within this study.

**Growth operators.** Take the vanilla decoder-only Transformer architecture as an example. $M$ is composed of $L$ decoder layers, each consisting of a multi-head self-attention sublayer and a feed-forward sublayer. Each decoder layer takes an input embedding that is presented as hidden states. As a result, there might be four Transformer dimensions to expand: layers, multi-head attention (mha), feed-forward network (ffn), and hidden states (hidden). We define the corresponding growth operators for these four dimensions as $\Phi = \{\phi_{layer}, \phi_{mha}, \phi_{ffn}, \phi_{hidden}\}$. Each operator $\phi \in \Phi$ initializes the extended parameters of the dimension randomly and reuses the weights from smaller models for the weights of larger models. According to (Karp et al., 2024), the behavior observed at initialization may not be a reliable indicator of final performance. Therefore, the influence of function-preserving is ignored. Appendix G contains more details about each growth operator.

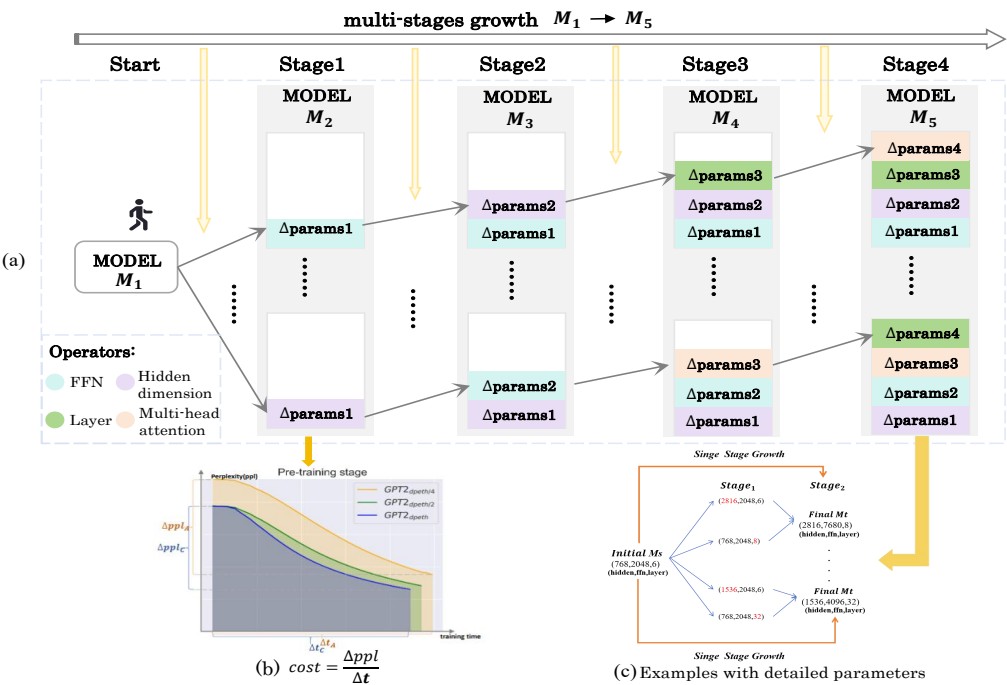

Figure 1: (a) Different growth paths for considering all the potential Transformer's growth dimensions. Each dimension's growth operator is executed during a single stage. There could be $A_4^4$ schedules by listing all potential growth operators' orders. (b) The relationship between ppl and training time affects the training efficiency of GPT2 with different depths. The ppl and training time values are from our experiments. (c) The 1B parameter target model may have various structures, including (2816, 7680, 8) and (1536, 4096, 32), which are arranged in a sequence with hidden states, ffn, and depth. It could grow from the initial model through both single-stage and two-stage growth in this example. Two-stage growth generates intermediate models such as (2816, 2048, 6) which increase the hidden states from 768 to 2816 during $Stage_1$.

**Multi-staged model growth.** Starting with a smaller model $M_1$, one or more growth operators $\phi \in \Phi$ can be employed to expand one or more dimensions, facilitating the growth of $M_1$ to a target model $M_k$, where the model parameter grows from $\theta_1$ to $\theta_k$. Traditional works (Chen et al., 2021; Wang et al., 2023) typically utilize one or more operators within a single stage to accomplish the goal, where $k = 1$ and no intermediary models arise. In the multi-stage scenario, the initial model $M_1$ increases by one dimension at each stage, ultimately achieving the target model $M_{k+1}$ following $k$ expansions. Figure 1 (a) provides examples of the Transformer's four potential dimensions for multi-stage growth. Figure 1 (c) presents examples that compare single-stage and multi-stage growth options. The sequence from $M_1$ to $M_{k+1}$ comprises multiple intermediary models, represented as $P = \{M_1, M_2, \ldots, M_{k+1}\}$, where the maximum value of $k$ is 4, determined by the complete set of growth operators. It is worth noting that for a given magnitude of parameters, there exist multiple distinct structures of the target model. Figure 1 (c) shows that the 1B model has different structures (2816, 7680, 8) and (1536, 4096, 32) in a sequence with hidden states, ffn, and depth.

## 3.2 TASK FORMULATION

A multi-stage model growth task can be formulated as below.

$$
\begin{aligned}
&Growth\ Path\ P : M_1(x, \theta_1) \Longrightarrow M_2(x, \theta_2) \Longrightarrow \cdots \Longrightarrow M_{k+1}(x, \theta_{k+1}) \\
&schedule : \bar{\epsilon} = \{\phi_1, \phi_2, \ldots, \phi_k\} \\
&operator : \theta_{k+1} = \phi_k(\theta_k), \phi_k \in \Phi
\end{aligned}
\tag{1}
$$

$\phi_k$ is a growth operator during one growth stage for expanding one dimension to grow the $M_k$ with $\theta_k$ to $M_{(k+1)}$ with $\theta_{k+1}$, and $\Phi$ is the growth operator set. $M_2, \ldots, M_k$ represent the intermediate models generated at the end of each growth stage, corresponding to the application of each growth operator aimed at increasing a specific dimension.

Each feasible $M_k$ structure possesses $A_4^4$ schedules by enumerating all possible $\Phi = \{\phi_{layer}, \phi_{mha}, \phi_{ffn}, \phi_{hidden}\}$ combinations. Each operator is restricted to a single use at each stage, thereby ignoring the occurrence of compound operator scenarios and reducing the schedule search space. Finding an optimal multi-staged growth path, which consists of multiple growth operators in a sequential order, to achieve better performance than other optional growth paths with limited training time is not trivial.

**Definition 1:** Given a computing budget of $C$ and the desired model parameter of $N$, an optimal training schedule identifies the optimum sequence of growth operators and intermediate model structures at each stage to use the least amount of computing power while maintaining target model performance.

## 3.3 MEASUREMENT OF OPTIMAL SCHEDULE USING MARGINAL UTILITY

To find an optimal schedule, we need to answer the first question: what is an appropriate measurement for an optimal schedule? We propose determining a good schedule measurement based on model performance and training costs. The model performance is evaluated using loss or perplexity (ppl) values, following most of the prior work for model optimization (Hoffmann et al., 2022). Figure 1(b) shows the relationship between ppl and training time, which can reflect the training efficiency of LLMs. It is non-trivial to find an appropriate objective function to optimize them simultaneously.

To address this problem, we borrow the concept of marginal utility in economics (Samuelson, 1937) and propose using the marginal utility of schedule (MUS) as the optimization objective. MUS evaluates the gains (reduction in ppl) that a model may obtain from an increase in cost (training time). Formally, MUS represents the derivation of the reduction of ppl to training time, which is calculated as:

$$
\sum_{k=1}^{4} \frac{\Delta ppl_{\phi_k}}{\Delta t(\phi_k)}, \phi_k \in \bar{\epsilon}
\tag{2}
$$

$\Delta ppl_{\phi_k} = ppl(M_k) - ppl(M_{k+1})$ is a positive value, representing the reduction in ppl achieved by the $M_{k+1}$ after training, when $\phi_k$ is selected as the operator for this expansion of $M_k$ to $M_{k+1}$. $\Delta t(\phi_k)$ means the training time from $M_k$ to $M_{k+1}$ costs.

Clearly, a higher benefit-cost ratio corresponds to a larger MUS. Given this MUS feature, we can shift our focus from finding an optimum schedule to establishing an optimal growth path that results in the highest MUS. A simple solution is to enumerate all candidate paths. Despite its simplicity and effectiveness, training all intermediate models and computing MUSs requires a significant cost. To address this, we investigate a learning-based method, SLOP, with restricted trial training.

## 3.4 SCHEDULE LEARNING VIA OPTIMAL PATH

We treat finding an optimum growth schedule as a dynamic programming problem, searching for the schedule in polynomial time. The goal is to find the optimal growth path from $M_1$ to $M_{k+1}$ with the highest MUS, as described in equation 2. Therefore, the objective of equation 2 can be transformed as below. By theoretically solving the dynamic programming problem, we could significantly reduce trial training costs, which include enumerating and assessing all feasible paths.

$$\arg\max_{\phi_k \in \bar{\epsilon}} \sum_{k=1}^{4} \frac{\Delta ppl_{\phi_k}}{\Delta t(\phi_k)} \iff \arg\max_{\phi_k \in \bar{\epsilon}} \sum_{k=1}^{4} \frac{\Delta ppl_{\phi_k}}{\Delta params(\phi_k)} \tag{3}$$

In equation 3, $\Delta params(\phi_k) = params(M_{k+1}) - params(M_k)$ represents the parameter that increases with each growth stage. Given that the natural logarithm, $\ln(\cdot)$, is a monotonically increasing function, equation equation 3 can be expressed as:

$$\arg\max_{\phi_k \in \bar{\epsilon}} \sum_{k=1}^{4} \frac{\Delta ppl_{\phi_k}}{\Delta params(\phi_k)} \iff \arg\max_{\phi_k \in \bar{\epsilon}} \sum_{k=1}^{4} \ln \frac{\Delta ppl_{\phi_k}}{\Delta params(\phi_k)}$$

$$\iff \arg\max_{\phi_k \in \bar{\epsilon}} \sum_{k=1}^{4} [\ln (\Delta ppl_{\phi_k}) - \ln (\Delta params(\phi_k))] \tag{4}$$

Optimizing the upper bound of equation 4 yields the re-formulation shown below (==$\iff$ represents the relaxation of solution space in this situation==):

$$\arg\max_{\phi_k \in \bar{\epsilon}} \sum_{k=1}^{4} [\ln (\Delta ppl_{\phi_k}) - \ln (\Delta params(\phi_k))] \iff$$

$$\arg\max_{\phi_k \in \bar{\epsilon}} \sum_{k=1}^{4} \ln (\Delta ppl_{\phi_k}) - \arg\min_{\phi_k \in \bar{\epsilon}} \sum_{k=1}^{4} \ln (\Delta params(\phi_k)) \tag{5}$$

Following a series of derivations (refer to proof), the objective function in equation 5 is transformed to:

$$\arg\max_{\phi_k \in \bar{\epsilon}} \underbrace{\frac{1}{D} \sum_{k=1}^{4} \log (q_{M_5}(x_i))}_{\text{①}} - \arg\min_{\phi_k \in \bar{\epsilon}} \underbrace{\sum_{k=1}^{4} \ln (\Delta params(\phi_k))}_{\text{②}} \tag{6}$$

where $D$ is the number of samples in the test set; $q_{M_5}(x_i)$ is the probability distribution predicted by the $M_5$ for any input $x_i$ in the test set.

**Proofs:** By utilizing just the first term of equation 5, we can get the following reformulation:

$$\arg\max_{\phi_k \in \bar{\epsilon}} \sum_{k=1}^{4} \ln (\Delta ppl_{\phi_k}) \iff \arg\max_{\phi_k \in \bar{\epsilon}} \ln (\sum_{k=1}^{4} \Delta ppl_{\phi_k})$$

$$= \arg\max_{\phi_k \in \bar{\epsilon}} \ln (ppl(M_1) - ppl(M_2) + ppl(M_2) - ppl(M_3) + \cdots + ppl(M_4) - ppl(M_5)) \tag{7}$$

$$= \arg\max_{\phi_k \in \bar{\epsilon}} \ln (ppl(M_1) - ppl(M_5))$$

The relaxation of the upper bound in equation 7 benefits from the monotonically increasing property of $\ln(\cdot)$. Since the initial $M_1$ before expansion is fixed, its corresponding ppl is a constant value, and we can obtain:

$$\arg\max_{\phi_k \in \bar{\epsilon}} \ln (ppl(M_1) - ppl(M_5)) \iff \arg\max_{\phi_k \in \bar{\epsilon}} [ppl(M_1) - ppl(M_5)]$$

$$\iff \arg\max_{\phi_k \in \bar{\epsilon}} ppl(M_1) - \arg\min_{\phi_k \in \bar{\epsilon}} ppl(M_5) \tag{8}$$

Consequently, the objective of the first term of equation 5 is to pursue $\arg\min_{\phi_k \in \bar{\epsilon}} ppl(M_5)$:

$$\arg\min_{\phi_k \in \bar{\epsilon}} ppl(M_5) \iff \arg\min_{\phi_k \in \bar{\epsilon}} \ln (ppl(M_5)) = \arg\min_{\phi_k \in \bar{\epsilon}} \ln (e^{\frac{1}{D} \sum_{i=1}^{D} EC_{M_5}(x_i)})$$

$$= \arg\min_{\phi_k \in \bar{\epsilon}} \frac{1}{D} \sum_{i=1}^{D} EC_{M_5}(x_i) = \arg\min_{\phi_k \in \bar{\epsilon}} \frac{1}{D} \sum_{i=1}^{D} [-P_{M_5}(x_i) \log(q_{M_5}(x_i))] \tag{9}$$

$$\iff \arg\max_{\phi_k \in \bar{\epsilon}} \frac{1}{D} \sum_{i=1}^{D} \log(q_{M_5}(x_i))$$

where $ppl(M_5) = e^{\frac{1}{D}\sum_{i=1}^{D} EC_{M_5}(x_i)}$, and $EC_{M_k}(x_i) = -P_{M_k}(x_i)\log(q_{M_k}(x_i))$ is used to calculate cross-entropy. $P_{M_k}(x_i)$ denotes the ground truth distribution, which is a constant value.

equation 6 splits the entire solution into two parts: 1) Finding the optimum target model with the highest average probability of accurate token prediction; 2) Enumerating all schedules and selecting the optimal one that satisfies equation 6.②.

In part.① of equation 6, for a model $M$, when the parameters $N$ and computation cost $C$ are fixed, the optimal loss can be predicted through the scaling law (Hoffmann et al., 2022), which is solely related to the model's parameters $N$ rather than its structure. Therefore, part.① in equation 6 can be taken as a constant. Then the optimization goal of equation 6 becomes:

$$
\begin{aligned}
\arg\min_{\phi_k \in \overline{\epsilon}} \sum_{k=1}^{4} \ln\left(\Delta params(\phi_k)\right) &= \arg\min_{\phi_k \in \overline{\epsilon}}[\ln\left(\Delta params(\phi_1)\right) + \cdots + \ln\left(\Delta params(\phi_4)\right)] \\
&= \arg\min_{\phi_k \in \overline{\epsilon}} \ln[\Delta params(\phi_1) * \cdots * \Delta params(\phi_4)]
\end{aligned}
$$
(10)

**Setup of optimal path.** Note that equation 10 has the same form as the objective function in the optimal path. To solve equation 10, given a directed graph $G = (V, E)$ where $V$ is the set of vertices and $E$ is the set of edges. Each edge $e_{ij} = (v_i, v_j)$ has a non-negative weight $w(v_i, v_j)$. The goal of equation 10 is to find a path from the source to the target vertex that meets certain conditions.

In this scenario, the vertices represent all of the potential intermediate model structures that could emerge as the model grows. The weights $w(v_i, v_j)$ of the edges in $E$ illustrate the variations in parameters at every stage of growth between the two vertices in $V$. Our objective is to find a path from the source vertex ($v_{source}^{M_1}$) to the destination vertex ($v^{M_5}$) in four stages, ensuring that the product of the edge weights is minimized:

$$
\min \prod_{k=1}^{4} w_k(v_i, v_j)
$$
(11)

Although the destination vertex is not unique, they share the same number of $M_5$'s parameters $N_5$. Formally, such constraints are defined as:

$$
\begin{aligned}
params(v^{M_5}) &= N_5, \\
\forall v^{M_5} \in V\_set_{target} &= (v_1^{M_5}, v_2^{M_5}, \ldots, v_t^{M_5})
\end{aligned}
$$
(12)

Now, we can use optimal path algorithms, such as the Dijkstra algorithm, to efficiently obtain the optimal schedule without trial training. Our algorithm details are shown in Algorithm 1, which outputs optimal schedules satisfying equation 2.

## 4 EXPERIMENTS

### 4.1 EXPERIMENTAL SETUP

**Datasets.** For pre-training, we use the redpajama (Computer, 2023) dataset and create 65B token training data using Llama's(Touvron et al., 2023a) training data mixture ratio. Among these, 25B tokens are used for initial model training and 10B tokens for model growth training at each stage.

**Downstream benchmarks.** For downstream task assessment, we use a set of common LLM evaluation benchmarks that include commonsense reasoning (PIQA (Bisk et al., 2020), Hellaswag (Zellers et al., 2019)), common aggregated benchmarks (BBH (Suzgun et al., 2022), Lambada (Paperno et al., 2016)), and math (GSM8K (Cobbe et al., 2021)).

**Model growth settings.** All of the models in our studies use vannila decode-only Transformer architectures. During model growth, we adhere to a few simple constraints contained in the existing LLM structure, as detailed in their published technical report, such as llama(Touvron et al., 2023a), qwen(Bai et al., 2023; Yang et al., 2024), baichuan(Yang et al., 2023), and mistral(Jiang et al., 2023). The constraints include: **1)** The hidden dimension size is a multiple of 128. **2)** The hidden dimension is either 8/3 or 4 times the ffn dimension. **3)** The number of attention heads should

---

**Algorithm 1** SLOP

---

**Input:** $G = (V, E)$, $v_{source}^{M_1}$, $V\_set_{target} = \{v_1^{M_5}, v_2^{M_5}, \dots, v_t^{M_5}\}$
**Output:** Traverse $v_t^{M_5} \in V\_set_{target}$, find the top K smallest $dist[v_t^{M_5}]$, and output its path (predecessor vertices) according to $prev[v_t^{M_5}]$

1: **Initialize:** Create vertex set $Q = \emptyset$
2: **for** each $v$ in $V$ **do**
3:  $dist[v] \leftarrow \infty$ //initial distance is set to infinity
4:  $prev[v] \leftarrow$ Undefined // Undefined $V$'s predecessor vertex
5:  Add $v$ to $Q$
6: **end for**
7: $dist[v_{source}^{M_1}] = 1$
8: **while** $Q \neq \emptyset$ **do**
9:  Find vertex $u$ satisfying $\min(dist) = dist[u]$
10:  Extract $u$ from $Q$: $Q = Q - u$
11: **end while**
12: **for** each adjacent vertex $v$ of $u$ **do**
13:  **if** $dist[v] > dist[u] * w(u, v)$ **then**
14:   $dist[v] \leftarrow dist[u] * w(u, v)$
15:   $prev[v] \leftarrow u$
16:  **end if**
17: **end for**

---

be divisible by the hidden dimension; nevertheless, this has no effect on the model's size. Check Appendix F for more details. With these constraints, we randomly select parameter N = 100M for the initial beginning point model with the structure (768, 2048, 6, 6), which corresponds to the four dimensions of (hidden, ffn, layer, and mha). Due to computational constraints, the target's N value is limited to 1B. The settings details for the growth operator are illustrated in Appendix G.

## 4.2 RESULTS AND ANALYSIS

Table 1: Training time for SLOP and other potential schedules on same training data over growth stages. Training time refers to the number of GPU hours required by the schedules to grow a model within the same data size. Since the attention head numbers do not lead to changes in parameters, we only consider the three dimensions (hidden-ffn-layer). To simplify the representation of schedule sequences, the abbreviations are used in table: l for layer, f for ffn, and h for hidden. SLOP utilizes minimal GPU time for training while maintaining superior performance in terms of perplexity.

| | Schedules | | | $Initial$ | $stage_1$ | $stage_2$ | $stage_3$ | **Sum** |
|---|---|---|---|---|---|---|---|---|
| Target structure | sequence | $\prod \Delta params$ | PPL | | Wall time (GPU hours) | | | |
| (2816,7680,8) | l-h-f | 6.59E+24 | 31.43 | 34.38 | layer 8.65 | hidden 27.53 | ffn 43.39 | 113.94 |
| | f-h-l | 4.14E+25 | 33.48 | 34.38 | ffn 11.21 | hidden 34.01 | layer 43.39 | 123.09 |
| | h-f-l | 1.81E+26 | 39.84 | 34.38 | hidden 22.12 | ffn 34.01 | layer 43.39 | 133.89 |
| (1280,3584,40) | h-f-l | 4.66E+24 | 31.1 | 34.38 | hidden 10.32 | ffn 11.80 | layer 45.22 | 101.72 |
| | l-h-f | 1.65E+26 | 38.65 | 34.38 | layer 19.66 | hidden 35.39 | ffn 45.22 | 134.66 |
| (1536,4096,32) | f-h-l | 4.24E+24 | 31.43 | 34.38 | ffn 9.14 | hidden 14.1 | layer 49.94 | 107.62 |
| | l-h-f | 1.52E+26 | 39.85 | 34.38 | layer 16.91 | hidden 37.36 | ffn 49.94 | 138.59 |
| (2560,6912,10) | f-l-h | 8.53E+24 | 32.99 | 34.38 | ffn 10.76 | layer 14.01 | hidden 44.40 | 103.55 |
| | h-f-l | 1.24E+26 | 38.65 | 34.38 | hidden 19.66 | ffn 29.00 | layer 44.40 | 127.44 |
| (2816,7680,8) | **SLOP** l-f-h | 1.69E+24 | **30.61** | 34.38 | layer 8.65 | ffn 12.97 | hidden 43.39 | **99.39** |

**Evaluate the theoretical validity of SLOP.** Table 1 shows a comparison of the pre-training performance of the SLOP recommended optimal schedule and other optional schedules. There may be multiple model structures for a 1B parameter model. The experiment includes all possible target model structures adhering to the previously stated structural constraints. For example, in Table 1,

one of the target structures is designated as (2816, 7680, 8), corresponding to the parameters (hidden, ffn, layer). We will apply this model structure notation uniformly throughout this section. The growth path ignores the expansion of attention head numbers, as it does not alter the parameters. The schedules in Table 1 cover all the possible sequences of the growth dimensions. Considering that the increased parameters at each stage are also part of the objectives for optimization, it is impractical to list all potential combinations of $\Delta params$ for each stage because of computational and time limitations. For each path, the minimal $\prod \Delta params$ are chosen as a representation, guaranteeing that the experiment includes all suboptimal possibilities.

We have the following observations from Table 1. SLOP demonstrates superior performance compared to alternative schedules in terms of perplexity, achieving a reduction in computational usage ranging from 2.74% to 35.48%. The outcomes of $stage_3$ further confirm the principle of the scaling law(Hoffmann et al., 2022), indicating that a model's performance is primarily influenced by its parameters and the training data while being less dependent on its structure.

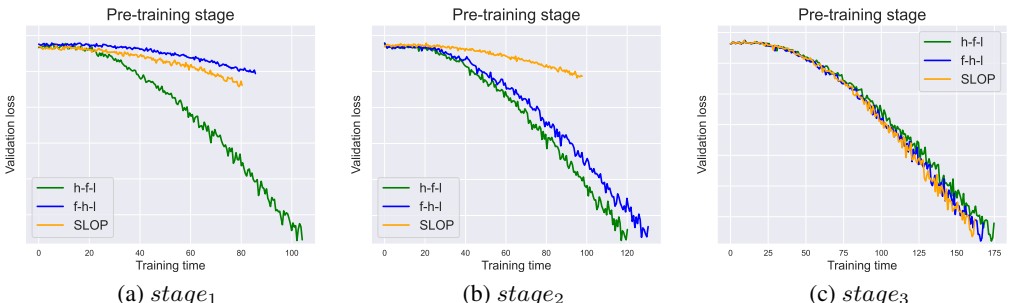

(a) $stage_1$      (b) $stage_2$      (c) $stage_3$

Figure 2: Validation loss vs. training time for different schedules during different growth stages.

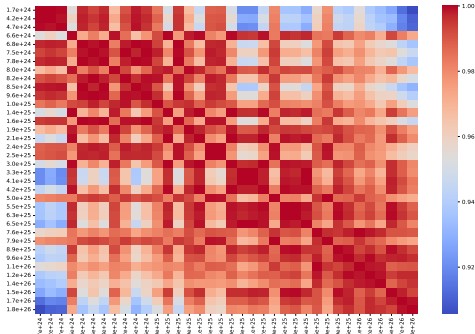

Figure 3: Correlation heatmap between different schedules

Figure 2 presents the loss results corresponding to different training schedules utilizing the target structure specified as (2816, 7680, 6, 8) in terms of (hidden, ffn, layer, head number). During the first growth stages, the intermediate model structure selected by SLOP fails to achieve optimal performance as a result of its comparatively limited parameters. Upon completion of the growth stages, SLOP demonstrates a significantly reduced training time compared to alternative schedules while preserving similar or superior loss values.

Additional experiments are being designed to further illustrate the capabilities of SLOP visually. We compute the correlation between the training times of various schedules and present the correlation heatmap shown in Figure 3. We can see a clear phase between these schedules. For a variety of schedules, the closer the $\prod \Delta params$ are, the higher the correlation of the training times. From a different point of view, this shows that the LLM training costs are positively related to $\prod \Delta params$. This supports our experimental setup that uses the minimal $\prod \Delta params$ to represent the potential growth sequence.

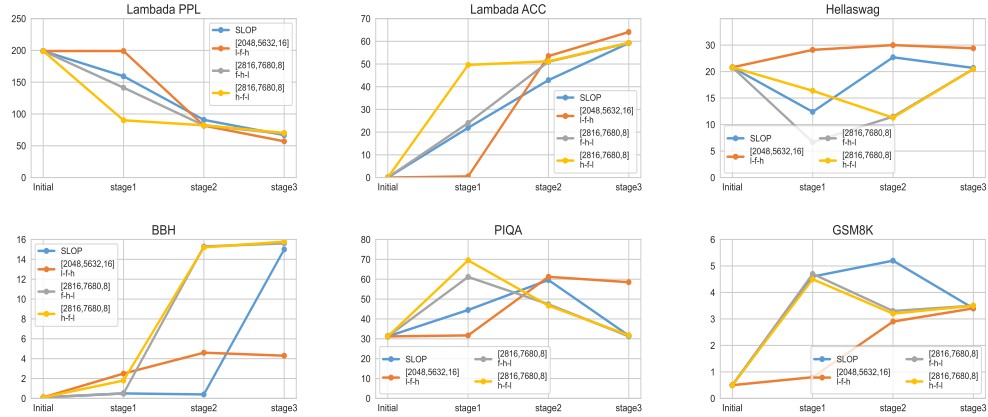

Figure 4: The performance of target models generated by various schedules following multi-stage model growth on representative downstream tasks. Blue line represents SLOP.

**Evaluate with other baselines.** We conduct a comparison of various growth schedule baselines to evaluate the effectiveness of SLOP, which are listed as follows. **1) SCHL-from scratch:** We train from scratch to obtain the target 1B parameter model using the previously specified 65b token pre-training dataset. **2) SCHL-single stage:** The initial model with 100M parameters is expanded to the target model of 1B parameters through a single-stage growth strategy. **3) SCHL-MSG:** The most recent proposed growth schedule by MSG (Yao et al., 2023) is also compared. **4) ELLE:** To assess the overall performance of model growth, we conduct a comparison with the model growth method ELLE, which allows for the incremental expansion of both the width and depth of the LLMs. Table 2 demonstrates that SLOP exhibits a reduction of 22.6% in training time when compared to SCHL-MSG and ELLE, while the perplexity remains nearly equivalent. Compared with SCHL-from scratch and SCHL-single stage, although the perplexity increased, SLOP saves 183.74% in training time.

**Performance on the downstream tasks.** As shown in Figure 4, we evaluate the target LLM's performance across a suite of popular benchmarks(Touvron et al., 2023b). It can be concluded that, in comparison to alternative schedules, SLOP demonstrates enhanced performance in most downstream tasks. This suggests that the knowledge gained through the optimal schedule during model growth can be effectively utilized for downstream tasks.

Table 2: Evaluation of perplexity and training time for SLOP compared to alternative baseline schedules, growing from identical initial models to target models.

| Target structure | Model | PPL | Wall time (GPU hours) |
|---|---|---|---|
| (2816,7680,8) | SCHL-from scratch | **26.43** | 282.01 |
| | SCHL-single stage | 28.76 | 207.92 |
| | SCHL-MSG | 31.18 | 101.96 |
| | ELLE | 30.6 | 121.86 |
| | **SLOP** | 30.61 | **99.39** |

### 4.3 ABLATION STUDY

**Effect of different initial models.** We evaluate the impact of different initial models for SLOP. The initial model $M_1$ consists of 450M parameters, structured as (1024, 4096, 16, 24), whereas the target model $M_5$ contains 1B parameters. The performance of different schedules, after multi-stage growth, is detailed in Table 3. The SLOP schedule offers significant computational savings while maintaining high performance, regardless of initial model changes.

Table 3: Training time for SLOP and other schedules on same data in the pre-training stage, utilizing a different initial model.

| | Schedules | | | $Initial$ | $stage_1$ | $stage_2$ | $stage_3$ | Sum |
|---|---|---|---|---|---|---|---|---|
| Target structure | sequence | $\prod \Delta params$ | PPL | Wall time (GPU hours) | | | | |
| | | | | | ffn | layer | hidden | |
| | f-l-h | 1.59E+17 | 35.08 | 102.67 | 8.55 | 28.02 | 44.61 | 183.85 |
| | | | | | layer | hidden | ffn | |
| (1152,3072,50) | l-h-f | 1.53E+26 | 35.48 | 102.67 | 23.10 | 37.24 | 44.61 | 207.62 |
| | | | | | ffn | hidden | layer | |
| | **SLOP** f-h-l | 4.76E+8 | 32.2 | 102.67 | 8.55 | 10.54 | 44.61 | **166.37** |
| | | | | | hidden | ffn | layer | |
| | h-f-l | 4.71E+24 | 33.01 | 102.67 | 9.66 | 11.87 | 45.71 | 169.92 |
| | | | | | layer | hidden | ffn | |
| (1152,4608,40) | l-h-f | 5.54E+25 | 34.02 | 102.67 | 19.66 | 29.49 | 45.71 | 197.54 |
| | | | | | ffn | hidden | layer | |
| | **SLOP** f-h-l | 2.28E+24 | 33.01 | 102.67 | 9.44 | 11.87 | 45.71 | **169.69** |

**Compatible to the special cases of two-dimensional expansion.** Existing studies on model growth often investigate expanding in depth and width dimensions (Yao et al., 2023; Yang et al., 2020; Shen et al., 2022; Wang et al., 2023). To validate the universality of SLOP, we limit the growth operators to $\psi = \{\varphi_{layer}, \varphi_{hidden}\}$. We set the structures of the initial model $M_1$ to (768, 2048, 6, 6), and the target model $M_5$ to (2816, 7680, 8, 22). Table 4 compares the performance of different schedules after two expansion stages for width and depth. The results presented the utility of SLOP in specific cases. The appendix provides more experimental details.

Table 4: Training time for SLOP and other schedules, growing with depth and width.

| | Schedules | | | $Initial$ | $stage_1$ | $stage_2$ | Sum |
|---|---|---|---|---|---|---|---|
| Target structure | sequence | $\prod \Delta params$ | PPL | Wall time (GPU hours) | | | |
| | | | | | hidden | layer | |
| (2816,7680,8) | h-l | 1.41E+17 | 38.65 | 34.38 | 68.03 | 43.39 | 145.8 |
| | | | | | layer | hidden | |
| | **SLOP** l-h | 1.38E+16 | 38.65 | 34.38 | 8.65 | 86.77 | **139.8** |

**The impact of multi-head attention.** Table 5 shows the effect of inserting $\varphi_{mha}$ on various positions within the SLOP recommended schedule (h-f-l, corresponding with Table 1). The target model structure is (1536, 4096, 32). Expanding the multi-head at $stage_1$ serves as the baseline. The placement of $\varphi_{mha}$ within the first three stages appears to have a small impact on training time. Positioning $\varphi_{mha}$ as the final stage will result in increased fluctuations in performance throughout the model growth process. We further investigate how varying the number of attention heads affects the target model's performance and downstream experiments, as detailed in Appendix C.2.

Table 5: The percentage savings in computing time pertains to the positioning of the $\varphi_{mha}$ at various growth stages in relation to the baseline, which involves expanding the $\varphi_{mha}$ at $stage_1$.

| **Schedules** | $stage_1$ | $stage_2$ | $stage_3$ | $stage_4$ | **Sum** |
|---|---|---|---|---|---|
| | computational savings (%) | | | | |
| f-head-h-l | 0.56 | -0.19 | -2.09 | -0.37 | -0.51 |
| f-h-head-l | 0.56 | -23.56 | 0.20 | 0.52 | -2.86 |
| f-h-l-head | 0.56 | -23.56 | -210.67 | -1.58 | -39.77 |

## 5 CONCLUSION

This study examines optimal model growth schedule learning problems, concentrating on determining a suitable sequence that integrates several operators to enhance performance for the target LLM. We present a cost-effective optimal path learning method within the framework of a multi-stage model growth scenario that could attain theoretically optimal results. Observe that we examine a straightforward scenario in which each growth dimension occurs just once along the path. We will leave more complex scenarios for future work.

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

## A   LIMITATIONS

In this work, we propose a theoretical solution for finding the optimal growth schedules for multi-stage growth involving all possible dimensions. However, we do not consider complex cases when multiple growth dimensions can combine at the same stage and execute more than once. Our work establishes a starting point for the development of self-adaptive growth schedules, vital to the efficient pre-training of LLMs. Another limitation is that, due to limited computing capacity and budget, the largest models in our experiments have 1 billion parameters, which is a significant difference from existing LLMs. This constraint is present in the vast majority of research projects, according to our knowledge.

## B   EXPERIMENTAL DETAILS

### B.1   DETAILS FOR THE EXPERIMENTS SET UP

**Datasets**: The training dataset mixture, comprising 65 billion tokens, adheres to the mixture ratio established by LLAMA(Touvron et al., 2023a), as detailed below:

Table 6: The pre-training data mixture ratio

| Dataset | ratio |
|---|---|
| CommonCrawl | 67.0% |
| C4 | 15.0% |
| Github | 4.5% |
| Wikipedia | 4.5% |
| Books | 4.5% |
| Arxiv | 2.5% |
| StackExchange | 2.0% |

**Tokenizer**: We tokenize the data with the byte pair encoding (BPE) algorithm (Sennrich et al., 2016), using the implementation from SentencePiece (Kudo & Richardson, 2018).

**Optimization**: Our models are trained using the AdamW optimizer, with the following hyper-parameters: $\beta_1 = 0.9$, $\beta_2 = 0.95$. We implement a cosine learning rate schedule, with the final learning rate set to 10% of the maximum value. Our weight decay is 0.1, and we apply gradient clipping at 1.0. We configure the batch size to 256K and use a warmup period of 2,000 steps. The details of the hyperparameters for our different models are given in Table 7.

Table 7: Model sizes, structures, and optimization hyper-parameters

| params | structure | learning rate | batch size | n tokens |
|---|---|---|---|---|
| 1B | $(d_{hidden}, d_{ffn}, N_{layer}, N_{head})$ | $3.0e^{-4}$ | 256K | 65B |

**Implementation**: Our code handles approximately 50K tokens per second per GPU on 2048 A100 GPUs with 80GB of RAM. Training for the final growth stage on our dataset, which includes 65B tokens, takes about 15 days in total.

### B.2   MORE DETAILS FOR THE GROWTH SCHEDULES IN THE MAIN EXPERIMENT

We present some sample schedule details (sch1, sch2, and sch3) for their respective expansion sequences, as shown in Table 8.

Table 8: Growth schedules expand in one dimension in each stage. The schedules listed are part of the main experiment used.

| Schedule | sequence | initial | stage1 | stage2 | stage3 |
|---|---|---|---|---|---|
| sch1 | h-f-l | (768,2048,6) | (2816,2048,6) | (2816,7680, 6) | (2816, 7680, 8) |
| sch2 | f-h-l | (768,2048,6) | (768,4096, 6) | (1536,4096, 6) | (1536, 4096, 32) |
| sch3 | l-f-h | (768,2048,6) | (768,2048,16) | (768,5632,16) | (2048,5632,16) |
| **SLOP** | l-f-h | (768,2048,6) | (768,2048,8) | (768,7680,8) | (2816,7680,8) |

## C  ABLATION STUDY

### C.1  THE IMPACT OF EXPANDING MHA IN DIFFERENT SEQUENCES

The experiments depicted in Figure 5 further back up our conclusion, demonstrating that the sequence of $\varphi_{mha}$ has minimal impact on the model's overall performance. In comparison, expanding the head as the final stage results in suboptimal performance in downstream tasks.

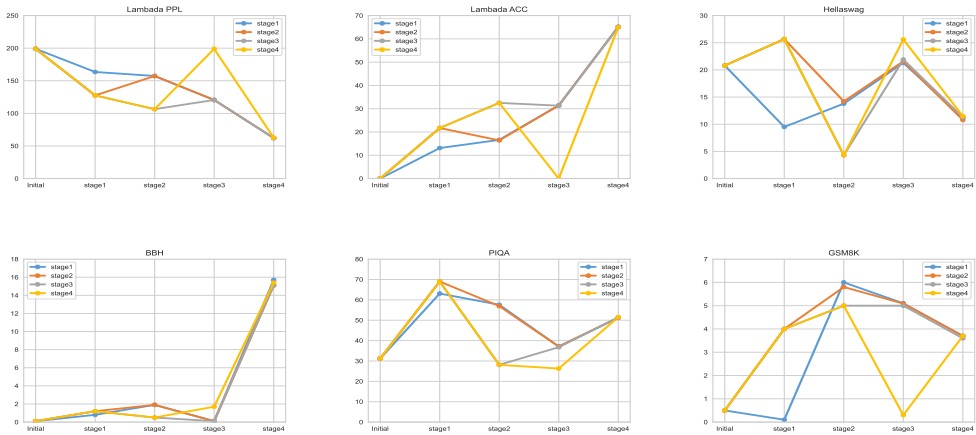

Figure 5: The target LLM's performance after multi-stage model growth pre-training on representative downstream tasks with $\varphi_{mha}$ in different sequences.

### C.2  THE IMPACT OF VARYING THE NUMBER OF HEADS.

Table 9 indicates a positive correlation between the number of heads and the increase in ppl.

Table 9: Perplexity of the target model with the head number increasing from 6 to 48 values, maintaining a fixed schedule of head-ffn-hidden-laye.

| | $stage_1$ | $stage_2$ | $stage_3$ | $stage_4$ | **Avg.** |
|---|---|---|---|---|---|
| head number | \multicolumn{5}{c}{Perplexity (PPL)} |
| $head_6$ | 107.53 | 104.87 | 70.48 | 31.29 | 78.54 |
| $head_8$ | 117.89 | 111.15 | 68.77 | 30.80 | 82.15 |
| $head_{12}$ | 117.87 | 110.25 | 70.27 | 30.61 | 82.25 |
| $head_{16}$ | 118.76 | 111.98 | 68.71 | 31.10 | 82.64 |
| $head_{24}$ | 120.30 | 116.81 | 68.69 | 31.64 | 84.41 |
| $head_{32}$ | 126.02 | 120.54 | 69.10 | 32.24 | 86.98 |
| $head_{48}$ | 126.40 | 126.02 | 71.17 | 32.99 | 89.15 |

We also investigate the influence of different numbers of attention heads on downstream task performance. As depicted in Figure 6, the optimal overall results are achieved with a head number of 48.

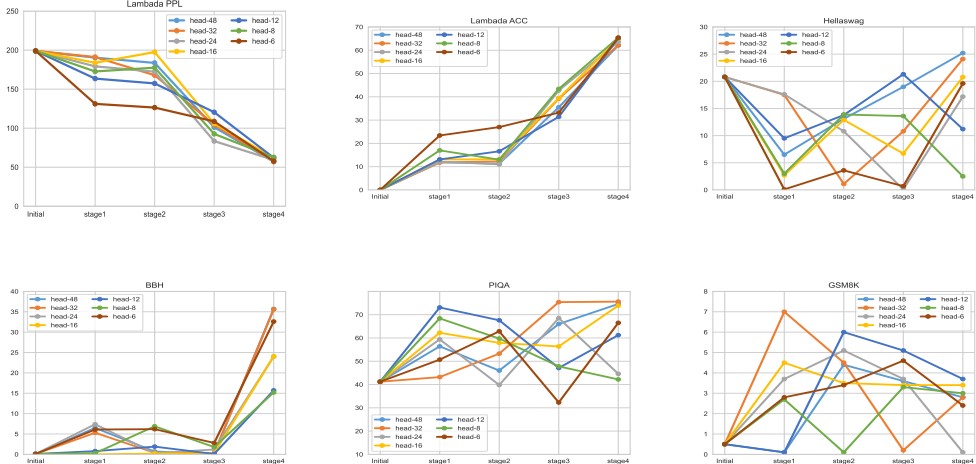

Figure 6: After multi-stage model growth, the performance of the target LLMs varies with different head numbers over representative downstream tasks.

## C.3 THE IMPACT OF DIFFERENT TARGET MODEL STRUCTURE.

To demonstrate the applicability of our method to all target structures, we conduct a supplemental experiment to evaluate the effectiveness of SLOP with an additional target structure (2048, 5632, 16), compared to baseline MSG. As shown in Table 10, the results further corroborate the versatility of SLOP across various target structures.

Table 10: Evaluation of perplexity and training time for SLOP compared to alternative baseline schedules, growing from identical initial models to target model (2048,5632,16).

| Target structure | Model | PPL | Wall time (GPU hours) |
|---|---|---|---|
| (2048, 5632, 16) | SCHL-single stage | **32** | 172 |
| | SCHL-MSG | 36 | 119 |
| | ELLE | 34 | 114 |
| | **SLOP** | 34 | **108** |

## C.4 THE IMPACT OF SMALLER PARAMETERS.

We further verify the generality of SLOP on smaller models, from 27M to 105M parameters. For a comparison, SLOP-105M and ELLE-105M employ different schedules and operators for model growth, progressively increasing from an initial model of 27M parameters with dimensions (384, 1024, 6) to a target model of 105M parameters with dimensions(768, 2048, 12). GPT-105M fixes the number of parameters at 105M and maintains a constant model size throughout each training stage. As presented in Table 11, the experimental results presented in the table demonstrate that our method is equally applicable to models with smaller parameter sizes.

Table 11: Evaluation of FLOPs and training time for SLOP compared to alternative baseline schedules in smaller models, growing from an initial model (384, 1024, 6) to a target model (768, 2048, 12).

| Target structure | Model | FLOPs(e18) | Wall time (GPU hours) |
|---|---|---|---|
| | GPT-105M | 4.46 | 3.10 |
| (768, 2048, 12) | ELLE-105M | 6.64 | 4.60 |
| | **SLOP-105M** | **4.08** | **2.83** |

## D  PERFORMANCE ON THE DOWNSTREAM TASKS

We conduct additional experiments comparing SLOP to the model growth baseline MSG, mentioned in Table 2, on some of the downstream tasks specified in Section 4.1 to assess SLOP's effectiveness. Table 12 below shows the results of the representative downstream tasks, demonstrating SLOP's robust performance on downstream tasks.

Table 12: The performance of SLOP compared to baseline MSG on representative downstream task.

| Target structure | Models | Lambada acc | Lambada ppl | BBH | Hellaswag |
|---|---|---|---|---|---|
| (2816,7680,8) | **SCHL-MSG** | 42.90 | 90.77 | 6.89 | **22.90** |
| | **SLOP** | **59.2** | **66.73** | **15.00** | 20.69 |

## E  CASES FOR THE GROWTH SCHEDULE PATHS

Figure 7 illustrates that the expansion of the Transformer model is configured with four dimensions: hidden_dim, ffn_dim, head_number, and layer_number. We select one dimension for each expansion, resulting in a total of four expansions. The number of parameters in the model expands from $a$ to $b$. For more complex scenarios in which each dimension can undergo multiple repetitions during the expansion process, such as hidden_dim, ffn_dim, head_number, hidden_dim, and layer_number, the entire search space becomes significantly larger and more complicated. In this paper, we omit this case, focusing only on expanding each dimension once along the expansion path.

As an example depicted in Figure 7, let's assume the initial model's four-dimensional parameters are set to (384, 1024, 6, 6), and through four rounds of expansion, the target parameter for the target model is achieved at 1B. If the target model's four-dimensional parameters are (768, 2048, 12, 12), we can calculate the number of possible path choices for the expansion schedule by multiplying the factorial of the number of steps in each dimension, which comes out to $4 * 3 * 2 * 1 = 24$. It is essential to recognize that various configurations exist for the structure of the target model. Another possible configuration is (1536, 4096, 32).

It is obviously impractical to traverse each schedule and select the final optimal one. Therefore, our goal is to transform this into an optimization problem of model metrics. Through a series of derivations, we can select the most suitable schedule before training to reduce the computational cost of achieving the optimal model.

## F  EXPLANATION FOR THE MODEL STRUCTURE CONSTRAINS

We acquire the structure of the LLaMA series and Qwen2 from their technical report, which is displayed in Table 13. The hidden dimension size of the LLaMA series (Touvron et al., 2023a;b; Dubey et al., 2024) and Qwen2 (Yang et al., 2024) is a multiple of 128. Furthermore, the hidden dimension of LLaMA1 65B is 8/3 of the ffn dimension. And the number of attention heads for all LLMs shown in Table 13 can be divided by the hidden dimension.

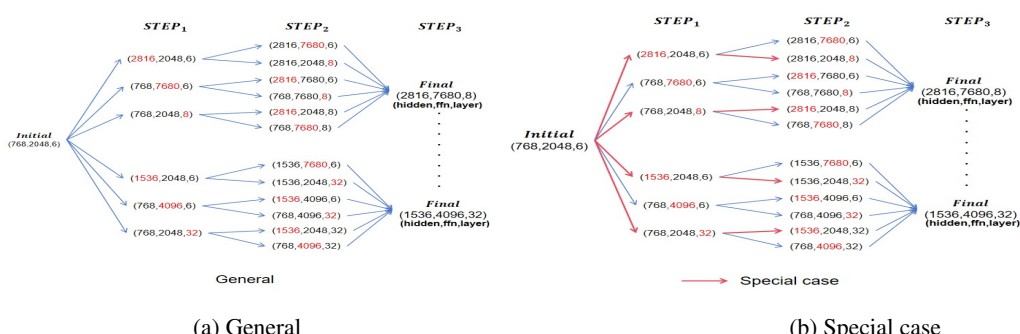

(a) General  (b) Special case

Figure 7: Examples for all the potential paths with special cases growth only width and depth.

Table 13: The model structure from different LLMs.

| Model | D_Hidden dimension | D_FFN | N_Heads | N_Layers |
|---|---|---|---|---|
| **LLaMA1 65B** | 8192 | 22016 | 64 | 80 |
| **LLaMA2 70B** | 8192 | 28672 | 64 | 80 |
| **LLaMA3 70B** | 8192 | 28672 | 64 | 80 |
| **Qwen2 72B** | 8192 | 29568 | 64 | 80 |

## G  THE GROWTH OPERATORS

In this study, we focus on the Transformer structure that is prevalent in existing LLMs. Transformer's possible growth dimensions are introduced below, while the operators of these dimensions are listed subsequently.

**Hidden states** $H^{l-1}$ represents the input for the Transformer layer $l$, which is a bi-dimensional tensor with $s$ and $h$ being the sequence and hidden dimension. When the $h$ changes, it affects every module of the Transformer structure. We overlook the position embedding in this work as it does not affect the expansion process. The hidden states are iteratively passing through the Transformer layers: $\underset{s \times h}{H^l} = Trans_l(\underset{s \times h}{H^{l-1}}), l \in [1, L]$, where L denotes the total number of the Transformer layers.

Each Transformer layer $l$ contains the modules that are important for the growth approach, which are described below:

**Multi-head attention (MHA)**: Multiple parallel self-attention heads make up MHA. The input $H$ of each layer is fed into the MHA mechanism, which can be formulated as follows:

$$\underset{s \times d}{K_i} / \underset{s \times d}{Q_i} / \underset{s \times d}{V_i} = \underset{s \times h}{H} \times \underset{h \times d}{W^{K/Q/V}_{head_i}}$$

$$\underset{s \times d}{H_{head_i}} = Attention(Q_i, K_i, V_i) = softmax(\frac{1}{\sqrt{d}} \times Q_i \times K_i^T) \times V_i \qquad (13)$$

$$\underset{s \times h}{H^{MHA}} = MHA(H) = [H_{head_1}, ..., H_{head_a}] \times \underset{(a \times d) \times h}{W^O}$$

where $H$ is applied to linear projection for generating queries, keys and values($Q/K/V$), utilizing different weights($W^{K/Q/V}$) for each transformation respectively. $H_{head_i}$ signifies the output of the i-th attention head with $a$ being the total number of heads. The output linear matrix $W^O$ generates the final result $H^{MHA}$, which is then delivered to the Feed-forward network.

**Feed-forward network (FFN)** is a Multi-Layer Perceptron responsible for applying a non-linear transformation to $H^{MHA}$ ($f$ is FFN's dimension of its internal representation):

$$\underset{s \times h}{H^{FFN}} = FFN(H^{MHA}) = GELU(\underset{s \times h}{H^{MHA}} \times \underset{h \times f}{W^{l_1}} + \underset{s \times f}{b^{l_1}}) \times \underset{f \times h}{W^{l_2}} + \underset{s \times h}{b^{l_2}} \qquad (14)$$

**MHA growth operator** $\varphi_{mha}$ refers to the act of introducing new heads within the multi-head attention module. As mentioned in Eq.13, the hyper-parameter $a$ controls the scaling of the multi-

head attention dimension. When the head number increases from $a_1$ to $a_2$, we keep the weights of the former heads fixed while assigning random values to the weights of the new heads.

$$W_i^{K/Q/V} = \begin{cases} W_i^{K/Q/V} & i \leq a_1 \\ random & a_i < 1 \leq a_2 \end{cases} \tag{15}$$

As the number of heads increases, alterations are also observed in the size of the corresponding weight matrix $W^O$ in Eq.13. We set the expanded portion of $W^O$ to be a random matrix $R$ as below:

$$\underset{(a_1 \times d) \times h}{W^O} \Rightarrow \underset{(a_2 \times d) \times h}{(W^O)'} = \begin{bmatrix} \underset{(a_1 \times d) \times h}{W^O} \\ \underset{((a_2 - a_1) \times d) \times h}{R} \end{bmatrix} \tag{16}$$

**FFN growth operator** $\varphi_{ffn}$ can be scaled up by increasing its internal representation's dimensionality. In Eq.14, the scaling of FFN expansion is controlled by the hyper-parameter $f$. Given a Transformer layer as an example, when the FFN's hidden dimension is increasing from $f_1$ to $f_2$, the extended part of $W^{l_1}$, $W^{l_2}$ and $b^{l_1}$ are initialized arbitrarily, written as $R$:

$$\underset{h \times f_1}{W^{l_1}} \Rightarrow \underset{h \times f_2}{(W^{l_1})'} = \begin{bmatrix} \underset{h \times f_1}{W^{l_1}} & \underset{h \times (f_2 - f_1)}{R^{W_{l_1}}} \end{bmatrix} \quad \underset{s \times f_1}{b^{l_1}} \Rightarrow \underset{s \times f_2}{(b^{l_1})'} = \begin{bmatrix} \underset{s \times f_1}{b^{l_1}} & \underset{s \times (f_2 - f_1)}{R^{W_{l_1}}} \end{bmatrix}$$

$$\underset{f_1 \times h}{W^{l_2}} \Rightarrow \underset{f_2 \times h}{(W^{l_2})'} = \begin{bmatrix} \underset{f_1 \times h}{W^{l_2}} \\ \underset{(f_2 - f_1) \times h}{R} \end{bmatrix} \tag{17}$$

**Hidden dimension growth operator** $\varphi_{hidden}$ is used to expand the dimension of the representation, which is originally sent into the Transformer layers. The scaling of hidden dimension expansion is controlled by the hyper-parameter $h$. When the hidden dimension of the representation is increasing from $h_1$ to $h_2$, we set the extended portion of $H$ to be random:

$$\underset{s \times h_1}{H} \Rightarrow \underset{s \times h_2}{H'} = \begin{bmatrix} \underset{s \times h_1}{H} & \underset{s \times (h_2 - h_1)}{R} \end{bmatrix} \tag{18}$$

Then each module in Transformer exhibits variations in the scaling for the parameters with hidden dimension expansion.

In the MHA module, we set the extended portion of $W^O$ to be random, and also the extended weight matrices of $K$, $Q$, and $V$ for each head are initialized randomly:

$$\underset{h_1 \times d}{W^{K/Q/V}} \Rightarrow \underset{h_2 \times d}{(W^{K/Q/V})'} = \begin{bmatrix} \underset{h_1 \times d}{W^{K/Q/V}} \\ \underset{(h_2 - h_1) \times d}{R} \end{bmatrix}$$

$$\underset{(a \times d) \times h_1}{W^O} \Rightarrow \underset{(a \times d) \times h_2}{(W^O)'} = \begin{bmatrix} \underset{(a \times d) \times h_1}{W^O} & \underset{(a \times d) \times (h_2 - h_1)}{R} \end{bmatrix} \tag{19}$$

In the FFN module, the extended portion of $W^{l_1}, W^{l_2}$ and $b^{l_2}$ are initialized randomly:

$$\underset{h_1 \times f}{W^{l_1}} \Rightarrow \underset{h_2 \times f}{(W^{l_1})'} = \begin{bmatrix} \underset{h_1 \times f}{W^{l_1}} \\ \underset{(h_2 - h_1) \times f}{R} \end{bmatrix} \tag{20}$$

$$\underset{f \times h_1}{W^{l_2}} \Rightarrow \underset{f \times h_2}{(W^{l_2})'} = \begin{bmatrix} \underset{f \times h_1}{W^{l_2}} & \underset{f \times (h_2 - h_1)}{N} \end{bmatrix} \quad \underset{s \times h_1}{b^{l_2}} \Rightarrow \underset{s \times h_2}{(b^{l_2})'} = \begin{bmatrix} \underset{s \times h_1}{b^{l_2}} & \underset{s \times (h_2 - h_1)}{R} \end{bmatrix}$$

**Layer operator** For the layer operator, we adopt the stacking method proposed in StackBERT (Deshpande et al., 2021).

