# OpenReview forum: "Model Growth Schedule learning via Optimal Path (SLOP) for Efficient LLM Pre-Training"
_ICLR.cc/2025/Conference — ICLR 2025 Conference Withdrawn Submission_

### Official Review · Reviewer_ksYf · 2024-10-30

**Soundness:** 2
**Presentation:** 2
**Contribution:** 2
**Rating:** 3
**Confidence:** 4

**Summary:**

To reduce the computational cost of pre-training LLMs, current methods start with a smaller model and gradually increase its size (e.g., by expanding the hidden size or adding layers) until reaching the target parameter count. This paper addresses how to design an optimal schedule for growing model size within this framework. The authors propose minimizing total marginal utility, specifically focusing on the overall decrease speed in perplexity across stages. After theoretical derivations, they simplify the problem to minimizing total parameter changes. Experimental results suggest that this approach reduces pre-training costs while achieving comparable or better PPL.

While the paper's motivation and approach are novel, there are several concerns regarding the formulation, theoretical derivations, and the main algorithm. In its current form, I hold a negative suggestion.

**Strengths:**

* This paper explores a significant question, presenting a clear motivation and novel perspective. The authors have conducted a thorough review of related work. Up to Section 3.2, the paper is generally well-written and easy to follow.

**Weaknesses:**

* **Formulation**. It is unclear why the authors limit the model to four stages and restrict each stage to only one growth operator. According to the objective in Eq.10, the cost decreases as the number of stages increases. Additionally, applying multiple growth operators in each stage is manageable—if there are four possible growth dimensions, there are only 15 compound operations in total, which is not prohibitively complex.

* **Derivation**. I have concerns about the derivation on Page 5. Beyond some notational issues, my main concern is with the equivalence in Eq.3 and Eq.5. I could not understand Eq.3, and the paper or appendix lacks an explanation. In Eq.5, $\arg\max$ yields a $\phi_k$, but the right side subtracts two $\phi_k$s. The subtraction is undefined. Furthermore, maximizing an upper bound does not necessarily optimize the original objective, making the relaxation in Eq.5 questionable.

I can hypothesize the authorss intended approach: starting with the RHS in Eq.3, applying a logarithmic function, and leveraging concavity. However, I still find Eq.3 unclear and would appreciate further clarification.

* **Algorithm**. Based on Figure 1, the optimal growth path resembles an application of the Viterbi algorithm, with complexity O(V+E) following the notation in line 286, which is lower than that of Algorithm 1. Additionally, Algorithm 1 may not be a dynamic programming approach, contrary to the claim in the abstract.

* **Experiments.** It is challenging to interpret the experimental results, particularly in Figures 2 and 3, where the x and y-axis values and meanings are unclear. While the authors state that they initialized a tiny model configuration randomly, it would be more rigorous to test alternative initial architectures of the same size. Otherwise, the results in Table 1 might appear cherry-picked.

In Table 2, the proposed method shows only marginal improvement over MSG, suggesting that the paper's contribution may be limited. The authors should also evaluate baseline models on downstream tasks, as shown in Figure 4.

**Questions:**

* Definition 1 is unclear. Given a compute budget, why is there a need to minimize compute power? What exactly is the variable in this problem—only the growth operator sequence, or does it also include training time for each stage?

* Please clarify why the number of stages is limited to 4.

* Please explain in detail why Eq.3 is valid.

* Does $\delta t$ represent wall time or GPU time?

* The use of \Leftrightarrow implies total equivalence, which does not seem to hold in Eq.5.

* Given the current formulation, where the search space is limited, why is Algorithm 1 necessary? Brute-force enumeration should be sufficient.

* In Table 1, there is an inconsistency between the calculated $\delta$ parameters and the actual GPU hours, particularly in the 4th and 7th rows. Could you explain this discrepancy?

* The authors state, "It is obviously impractical to traverse each schedule and select the final optimal one." Could you provide a concrete example of the search space (i.e., |V| and |E|) to demonstrate why this is impractical?

---

> ### Author Response · Authors · 2024-11-22
>
> We first want to thank the reviewer for their thorough review and largely positive comments. In particular, they highlight that the method is novel, intuitive, well-formulated, situated wrt related work, and has strong experimental results.
> In the rest of this response we will address the weaknesses and questions raised in the review.
>
> ---
>
> **Q1**: It is unclear why the authors limit the model to four stages and restrict each stage to only one growth operator.
>
> **A1**: This is a great point. In line with commonly used model growth methods`[1, 2]`, we restrict SLOP to expanding a single dimension at each stage. For the Transformer structure, there could be four potential dimensions for expansion: hidden_dim, head_num, ffn_dim, and layer. Therefore, this work involves a maximum of four stages.
>
> There could be more complex cases where multiple growth dimensions can combine at the same stage and execute more than once, as mentioned in the Limitation section. It would be interesting to explore the more complex cases. However, we believe the situations should also adhere to some constraints that existing LLMs (such as llama`[3]`, qwen`[4]`, baichuan`[5]`, and mistral`[6]`) often comply with, which may be mainly due to the GPU parallel strategies. These constraints include:
> 1. The hidden dimension is either 8/3 or 4 times the ffn dimension.
> 2. The number of attention heads should be divisible by the hidden dimension;
>
> We'll leave this for future work.
>
> >_[1] Gesmundo, Andrea, and Kaitlin Maile. "Composable function-preserving expansions for transformer architectures." arXiv preprint arXiv:2308.06103 (2023)._
>
> >_[2] Yao, Yiqun, et al. "2x faster language model pre-training via masked structural growth." arXiv preprint arXiv:2305.02869 (2023)._
>
> >_[3] Touvron, Hugo, et al. "Llama: Open and efficient foundation language models." arXiv preprint arXiv:2302.13971 (2023)._
>
> >_[4] Yang, An, et al. "Qwen2 technical report." arXiv preprint arXiv:2407.10671 (2024)._
>
> >_[5] Yang, Aiyuan, et al. "Baichuan 2: Open large-scale language models." arXiv preprint arXiv:2309.10305 (2023)._
>
> > _[6] Jiang, Albert Q., et al. "Mistral 7B." arXiv preprint arXiv:2310.06825 (2023)._
>
> --
>
> **Q2**: Applying multiple growth operators in each stage is manageable—if there are four possible growth dimensions, there are only 15 compound operations in total, which is not prohibitively complex.
>
> **A2**: Thank you for your informative query. For a target model with a parameter count of 1B, under the constraints mentioned in Section 4 Model Growth Settings, there exist multiple combinations of four dimensions: (1280, 3584, 10, 40), (1536, 4096, 12, 32), (1792, 4864, 14, 20), (2048, 5632, 16, 16), (2304, 6144, 18, 12), (2560, 6912, 20, 10), and (2816, 7680, 22, 8). Consequently, the theoretical size of the search space is $7*A_4^4 = 168$ if the model requires expansion across four dimensions in a four-stage expansion process, with each stage involving the expansion of only one dimension. We only present a few representative results in the paper. Our method, SLOP, requires no training and can identify the optimal schedule path among these 168 paths. Therefore, we consider SLOP valuable for optimizing model growth.
>
> ---
>
> **Q3**: In Table 2, the proposed method shows only marginal improvement over MSG, suggesting that the paper's contribution may be limited. The authors should also evaluate baseline models on downstream tasks, as shown in Figure 4.
>
> **A3**: Due to time restrictions, we conducted additional experiments comparing SLOP to the model growth baseline MSG on some of the downstream tasks specified in the manuscript to assess SLOP's effectiveness. The table below shows the results of the experiments, demonstrating SLOP's robust performance on downstream tasks.
>
> | Target structure | Models| Lambada acc | Lambada ppl | BBH | Hellaswag|
> | --------:   | :----:   | :----:  | :----:   | :----:  | :----:  |
> |(2816,7680,8)|SCHL-MSG|42.9|90.77|6.89|22.9|
> |(2816,7680,8)|SLOP|**59.2**|**66.73**|**15.00**|20.69|
>
> ---
>
> **Q4**: Definition 1 is unclear. Given a compute budget, why is there a need to minimize compute power? What exactly is the variable in this problem—only the growth operator sequence, or does it also include training time for each stage?
>
> **A4**: The variable is the growth operator sequence, referred to as growth schedules. Since the value of training time is directly related to growth schedules, training time is considered as the dependent variable of growth schedules. Given that training time can be clearly quantified during the model training process, we chose training time reduction as one of the marginal utility optimization objectives.
>
> ---

---

> ### Author Response · Authors · 2024-11-22
>
> **Q5**: Please explain in detail why Eq.3 is valid.
>
> **A5**： Thank you for bringing up this important clarification point. We omit some of the reasoning steps for brevity in the paper. Given a fixed computing budget, there exists a positive correlation between the GPU time required for training and the FLOPs. Therefore, it can be stated that: $\Delta t = f(\Delta FLOPs)$. Based on the theory of scaling laws`[1]`: $FLOPs \approx 6ND$, where $N$ represents model size(params) and $D$ denotes the number of training tokens. When the training dataset remains unchanged, $D$ is a constant value. Therefore, $\Delta t = f(\Delta FLOPs) \approx g(\Delta params)$, while $t$ and $params$ exhibit the same trend of increase, we can conclude that:
>
> $$ \mathop{argmax}\limits_{\phi_k \in \overline{\epsilon}}{\sum_{k=1}^4 \frac{\Delta ppl_{\phi_k}}{\Delta t(\phi_k)}} \Longleftrightarrow \mathop{argmax}\limits_{\phi_k \in \overline{\epsilon}}{\sum_{k=1}^4 \frac{\Delta ppl_{\phi_k}}{\Delta params(\phi_k)}}$$
>
> > _[1] Kaplan, Jared, et al. "Scaling laws for neural language models." arXiv preprint arXiv:2001.08361 (2020)._
>
> ---
>
> **Q6**: Does δt represent wall time or GPU time?
>
> **A6**: δt represents GPU time.
>
> ---
>
> **Q7**: The use of $\Leftrightarrow$ implies total equivalence, which does not seem to hold in Eq.5.
>
> **A7**: Thank you for pointing out the issue for us. Equation 5 is based on the following proof and supporting evidence:
>
> Existing sets $A,B$, and $A-B = \\{a-b:a \in A, b \in B \\}$. Suppose $max(A)$ and  $max(B)$, show that $max(A-B)$ also exists and that: $max(A-B) \geq max(A) - min(B)$
>
> Prove: Since $max(A) \in A$ and  $max(B) \in B$, the definition of $A-B$ that $max(A-B) \in A-B $ holds. Now let $x \in A-B \Rightarrow $ there exists $a \in A, b \in B$ such that $x=a-b$.
>
> According to the definition of max and min, $max(A) \geq a$, $min(B) \leq b$ concludes $-min(B) \geq -b$. Therefore $max(A)-min(B) \geq a-b =x \in A-B$ holds for all $x \in A-B$.
>
> The solution space of $max(A)-min(B)$ includes the solution of $max(A-B)$, thus we relax $max(A-B)$ to $max(A)-min(B)$, and the notation should be **$\Rightarrow$**.
>
> Additionally, the relaxation in our Equation 5 follows the proofs of Equation 3 in `[1]`.
>
> > _[1] Xu, Jingjing, et al. "Vocabulary learning via optimal transport for neural machine translation." arXiv preprint arXiv:2012.15671 (2020)._
>
> ---
>
> **Q8**: Given the current formulation, where the search space is limited, why is Algorithm 1 necessary? Brute-force enumeration should be sufficient.
>
> **A8**: As elucidated in response to Question 2, the theoretical size of the search space is 7*A44=168 if the model requires expansion across four dimensions in a four-stage expansion process, with each stage involving the expansion of only one dimension. Furthermore,the current method could be compatible with finer-grained growth phases (e.g., each dimension could be expanded more than once, as we discussed in Appendix A.), in which cases the search space could increase multiple times.  Moreover, when model size increases, particularly for models with more than 10 billion parameters, the search space expands dramatically. These are the issues we will explore in our future work. Therefore, we consider Algorithm 1 to be an effective approach. Of course, other algorithms that identify optimal paths are equally applicable.
>
> ---

---

> ### Author Response · Authors · 2024-11-22
>
> **Q9**: In Table 1, there is an inconsistency between the calculated δ parameters and the actual GPU hours, particularly in the 4th and 7th rows. Could you explain this discrepancy?
>
> **A9**: This discrepancy is due to the calculation method of GPU time, which is based on the sum of FLOPs in current stage, and FLOPs are positively correlated with the number of model parameters in each stage. When calculating, there may arise a situation where the product becomes larger as the difference between two numbers decreases: $|a1-b1| <  |a2-b2|$, $a1 \ast b1>a2 \ast b2$.
>
> The following table illustrates this discrepancy through an example. For the sake of computational convenience, we assume that when the number of parameters increases by 1, the corresponding FLOPs increase by α. For Structure 1, assuming the original number of parameters is K, with an increase of 3 in the first stage and an additional increase of 5 in the second stage, the total FLOPs after the two stages would be calculated as follows: $K\alpha + (K + 3)\alpha + (K + 3 + 5)\alpha = (3K + 11)\alpha$. Therefore, the increase in FLOPs is $11\alpha$. Similarly, for Structure 2, the increase in FLOPs can be derived as $16\alpha$. Then, under a constant GPU utilization, the GPU time for Structure 1 is less than that for Structure 2. However, in terms of $\prod \Delta params$, Structure 1 has a product of $3 \ast 5=15$, while Structure 2 has a product of $7 \ast 2=14$.
>
> | Target structure | $\prod \Delta params$| Added params in stage1 | Added params in stage2 | Sum of FLOPs added in two stages |
> | --------:   | :----:   | :----:  | :----:   | :----:  |
> |Structure 1|15|3|5|$11\alpha$|
> |Structure 2|14|7|2|$16\alpha$|
>
> ---
>
> **Q10**: The authors state, "It is obviously impractical to traverse each schedule and select the final optimal one." Could you provide a concrete example of the search space (i.e., |V| and |E|) to demonstrate why this is impractical?
>
> **A10**: Thank you for giving us the opportunity to elaborate. "traverse each schedule and select the final optimal one" refers to the process of obtaining the optimal schedule: Initially, we must enumerate every possible combination of dimensions. Subsequently, for each schedule, we sequentially expand and fully train the model, obtaining the best schedule that has the optimal performance and training time ratio.
>
> The above-mentioned process is impractical to traverse because its time and computation are costly.Especially for the large size models. For instance, the technical report of Llama2`[1]` notes that an increase in model parameters leads to a corresponding increase in the GPU time required for training each model, with a 7B model requiring 184,320 GPU hours. Assuming the target model aims for the 7B parameter, there are 62 possible target models (under the constraints in Answer 1), hence the time required to train all possible schedules on a four-stage model growth process is greater than $62 \ast A_4^4 \ast 184,320= 274,268,160$ hours(|V|=4031, |E|=4030). In such circumstances, obtaining the optimal schedule through exhaustive search and training is evidently impractical.
>
> Our method, however, does not require training the model to find an optimal schedule. Instead, it determines the target model architecture and growth operator sequences(schedules) berfore pre-training, striking a balance between the performance of the target model and the training time.
>
> > _[1] Touvron H, Martin L, Stone K, et al. Llama 2: Open foundation and fine-tuned chat models[J]. arXiv preprint arXiv:2307.09288, 2023._
>
> ---

---

> > ### Comment · Reviewer_ksYf · 2024-11-23
> > **Thank you for the response.**
> >
> > I appreciate the authors' detailed responses; however, I find them unhelpful in addressing my concerns. In summary, all my concerns listed in the Weaknesses section, as well as questions 2, 3, 5, 6, and 8 from the Questions section, remain unaddressed.
> >
> > Response A1 essentially reiterates content already presented in the paper without addressing my question. Let me clarify. For instance, when training a 100B LLM, it is more practical to scale the model incrementally, e.g., 100M ->1B -> 10B -> 100B, while optimizing the end-to-end training cost. The method proposed in this paper optimizes the cost of each individual stage but does not account for the overall training process. Although compound growth operators could potentially resolve this limitation, the authors have not provided a rationale for why this approach is infeasible, which diminishes the validity of the paper.
> >
> > Response A2 includes an enumeration, which I appreciate, but let me expand on the compound operator case. In each stage, there are at most 7 model configurations, with up to 15 compound operators applicable to each configuration. This gives |V| = 7 and |E| = 15 per stage. With a maximum of 4 stages, the full transition graph has at most 5*|V| = 35 nodes and 4*|E| = 60 edges. This graph size is manageable for algorithms like DP or Dijkstra. However, Response A2 does not explain why compound operators were not considered. Similar concerns apply to Q8/A8.
> >
> > I appreciate the additional results provided in Response A3, but I strongly recommend a thorough evaluation across all 6 benchmarks for comprehensive validation.
> >
> > Regarding Q4/A4, the phrasing of Definition 1 should exclude the term "use the least amount of computing power" for greater clarity and correctness.
> >
> > The explanation in Response A5 is unconvincing. From my perspective, the assumption that models with the same number of parameters should perform differently invalidates the scaling law argument. Otherwise, it would not be necessary to analyze the target structure (or model configuration) presented in Table 1. This weakens the justification provided.
> >
> > In relation to Q7/A7, the lemma in the response is fundamentally flawed. There are two errors in Eq.5: (1) the RHS represents an upper bound of the LHS, so it does not establish equivalence; (2) the RHS should be \arg(\max−\min), not argmax−argmin. Additionally, the response fails to provide any new or helpful insights.
> >
> > For Q9, why are training hours considered when your method does not involve any actual training? This seems to inflate the reported numbers unnecessarily. The total number of enumerations is only 1488, which is entirely feasible for modern computational resources.

---

> > > ### Author Response · Authors · 2024-11-23
> > >
> > > We sincerely thank you very much for the evaluation of our response and feedback.
> > >
> > >
> > > Q:Response A1 essentially reiterates content already presented in the paper without addressing my question. Let me clarify. For instance, when training a 100B LLM, it is more practical to scale the model incrementally, e.g., 100M ->1B -> 10B -> 100B, while optimizing the end-to-end training cost. The method proposed in this paper optimizes the cost of each individual stage but does not account for the overall training process. Although compound growth operators could potentially resolve this limitation, the authors have not provided a rationale for why this approach is infeasible, which diminishes the validity of the paper.
> > >
> > > A: We did not investigate scaling the model progressively because it is outside the scope of this work. We believe that the scheduling for LLM scaling is a complicated problem that involves a range of various conditions, and we are not able to handle them all in one work.
> > >
> > > Q：Response A2 includes an enumeration, which I appreciate, but let me expand on the compound operator case. In each stage, there are at most 7 model configurations, with up to 15 compound operators applicable to each configuration. This gives |V| = 7 and |E| = 15 per stage. With a maximum of 4 stages, the full transition graph has at most 5*|V| = 35 nodes and 4*|E| = 60 edges. This graph size is manageable for algorithms like DP or Dijkstra. However, Response A2 does not explain why compound operators were not considered. Similar concerns apply to Q8/A8.
> > >
> > > A:We have clarified that the compound operators are beyond the scope of this paper and are part of our ongoing work.
> > >
> > >
> > > Q:I appreciate the additional results provided in Response A3, but I strongly recommend a thorough evaluation across all 6 benchmarks for comprehensive validation.
> > >
> > > A: We'll attach all six benchmarks later. Some of them have not finished.
> > >
> > > Q:Regarding Q4/A4, the phrasing of Definition 1 should exclude the term "use the least amount of computing power" for greater clarity and correctness.
> > >
> > > A: Thanks for your valuable comments. The revision is listed as below:
> > > Definition 1: Given a computing budget of C and the desired model parameter of N, an optimal training schedule identifies the optimum sequence of growth operators and intermediate model structures at each stage while maintaining target model performance.
> > >
> > >
> > > Q:The explanation in Response A5 is unconvincing. From my perspective, the assumption that models with the same number of parameters should perform differently invalidates the scaling law argument. Otherwise, it would not be necessary to analyze the target structure (or model configuration) presented in Table 1. This weakens the justification provided.
> > >
> > > A:We didn't expect the same number of parameters to perform differently. But according to openAI's scaling law results, the structure of models has little impact on the performance. But different structures may have different training costs.
> > >
> > > Q:In relation to Q7/A7, the lemma in the response is fundamentally flawed. There are two errors in Eq.5: (1) the RHS represents an upper bound of the LHS, so it does not establish equivalence; (2) the RHS should be \arg(\max−\min), not argmax−argmin. Additionally, the response fails to provide any new or helpful insights.
> > >
> > > A:We formulate Eq. 5 based on the work of [1], which is recognized as an ACL best paper.
> > >
> > > > _[1] Xu, Jingjing, et al. "Vocabulary learning via optimal transport for neural machine translation." arXiv preprint arXiv:2012.15671 (2020)_
> > >
> > > Q:For Q9, why are training hours considered when your method does not involve any actual training? This seems to inflate the reported numbers unnecessarily. The total number of enumerations is only 1488, which is entirely feasible for modern computational resources.
> > >
> > > A:Table 1 shows the results of actual training, the target models listed in the table are trained, and the correctness of our method is verified by the actual training results. We can't agree that all 1,488 models need to be enumerated. It's a great drain on resources and money. Our method provides a way to save them.

---

> > > > ### Comment · Reviewer_ksYf · 2024-11-24
> > > > **Your responses are not convincing enough.**
> > > >
> > > > I appreciate the authors' prompt response, but unfortunately, I do not find the responses satisfactory in addressing my questions.
> > > >
> > > > **I feel that the authors' response lacks sufficient scientific rigor.** Regarding Eq. 5, I pointed out that your lemma and proof in the rebuttal are incorrect. The \(\geq\) operator should instead be \(\leq\) in the conclusion. **I am quite surprised that the authors did not recognize this basic mathematical error and instead dealed with my concern with a reference to a "best paper."** The correct derivation of Eq. 5 does not rely on any "best paper" references — it is as simple as:
> > > > $$
> > > > \max_x f(x) - g(x) \leq \max_x f(x) - \min_x g(x).
> > > > $$
> > > > This demonstrates that Eq. 5 in the paper is optimizing an upper bound, not a lower bound. First, this is an approximation and should not use the equivalence operator $\Leftrightarrow$. Second, optimizing the upper bound does not inherently optimize the original objective. Finally, the paper does not mention "lower bound" or "upper bound" distinctions, making the entire derivation unjustified.
> > > >
> > > > For reference, Eq. 3 in the cited "best paper" transforms the equation to \(\max_x f(x) - \max_x g(x)\), which provides a valid lower bound. The authors should have recognized this distinction.
> > > >
> > > > Without incorporating multi-stage growth and compound operators, the current problem formulation exhibits significant limitations and may not be practically useful. These elements do not drastically increase problem complexity, and I encourage the authors to amend these shortcomings in future submissions.
> > > >
> > > > Thank you for the additional explanation regarding A5. However, I would like to engage in a deeper discussion with the authors. First, how do you define $\Delta t$? Does it represent the time to train the model until convergence, or is it for a fixed wall time? Second, the scaling law specifies the FLOPs for training the model per step, not until convergence. Why, then, does Eq.3 hold? The answers to these questions are largely absent in the current version of the paper, which remains unchanged after the rebuttal phase.
> > > >
> > > > Regarding Q9, I regret to say that the authors' response completely diverges from my original question. I was not referring to experiments or evaluations but rather to the search process itself. The authors repeatedly stated in the rebuttal:
> > > > "Our method, SLOP, requires no training and can identify the optimal schedule path among these 168 paths."
> > > > If this is the case, then training is not necessary to verify the method’s correctness during the search process, right? My question was: given that the search space (168 or 1488 paths) is relatively small, why is a tailored algorithm, as described in Sec. 3.4, required? A brute-force approach should suffice, significantly diminishing the value of the proposed method.

---

> > > > > ### Author Response · Authors · 2024-11-26
> > > > > **Summary of your concerns**
> > > > >
> > > > > Thanks for reviewer’s detailed reply. We’d like to summarize the whole discussion. Hope we can minimize our differences.
> > > > >
> > > > > ---
> > > > >
> > > > > **1. The problem Equation 5, we summarized the whole discussion as shown below; could you check that this addresses the issues?**
> > > > >
> > > > >  In the **Eq 3’s Proofs of ACL paper** VOLT`[1]` that we referenced, the following theory was utilized:
> > > > > $$min(f(x)-g(x)) \geq min(f(x))-max(g(x))$$
> > > > >
> > > > > The right side serves as the **lower bound** for the left side, and therefore, the lower bound inequality is approximately relaxed to:
> > > > > $$min(f(x)-g(x)) \Leftrightarrow min(f(x))-max(g(x))$$
> > > > >
> > > > > In our paper, the following theory is utilized:
> > > > > $$max(f(x)-g(x)) \leq max(f(x))-min(g(x))$$
> > > > >
> > > > > The right side serves as the **upper bound** for the left side, and therefore,  we relax the upper bound inequality approximately as to:
> > > > > $$max(f(x)-g(x)) \Leftrightarrow max(f(x))-min(g(x))$$
> > > > >
> > > > > To argue this issue from a different perspective, $min(f(x)-g(x)) \Leftrightarrow min(f(x))-max(g(x))$  can be written as:
> > > > > $$-min((-f(x))-(-g(x))) \Leftrightarrow -min(-f(x))-(-max(-g(x)))$$
> > > > > $$-min(f^{'}(x)-g^{'}(x)) \Leftrightarrow -min(f^{'}(x))-(-max(g^{'}(x)))$$
> > > > > $$max(f^{'}(x)-g^{'}(x)) \Leftrightarrow max(f^{'}(x))-min(g^{'}(x))$$
> > > > >
> > > > > Thus, the relaxation $max(f(x)-g(x)) \Leftrightarrow max(f(x))-min(g(x))$ holds.
> > > > >
> > > > > Thank the reviewer for pointing out the issue with the notation. We have followed the VOLT paper in using the symbol $\Leftrightarrow$ to represent relaxation, which is not a rigorous usage. We will revise the notation in the revision to clearly indicate that it represents relaxation.
> > > > >
> > > > > > _[1] Xu, Jingjing, et al. "Vocabulary learning via optimal transport for neural machine translation." arXiv preprint arXiv:2012.15671 (2020)_
> > > > >
> > > > > ---
> > > > >
> > > > > **2.Regarding the more complicated scenarios such as incrementally increasing the model size and end to end training.**
> > > > >
> > > > > We are unable to address these concerns because they fall outside the scope of this study, as indicated in the limitations and during our discussion. Of course, the best solution to all problems is to use just one method. However, due to the complexity of practical application scenarios, they are not just math problems (as listed in the previous replies) and frequently require establishing corresponding methods for each scenario. We believe that scheduling for LLM scaling is a complex challenge involving a number of different circumstances (such as the LLM structure constraints) that need to be resolved with a series of works.
> > > > >
> > > > > ---
> > > > >
> > > > > **3. We can’t agree with that he optimization algorithm is not necessary since it could be enumerate. Especially it’s not convincing “The total number of enumerations is only 1488, which is entirely feasible for modern computational resources”. As we have analyzed below, it is significantly consuming GPU resources.**
> > > > >
> > > > > Regarding the question of whether optimization is necessary, we regret that we cannot agree with your viewpoint. The significance of our algorithm does not lie in the choice of the shortest path algorithm, as there can be numerous shortest path algorithms, such as brute-force enumeration, Dijkstra, and others. $Algorithm_1$ is merely an example of one such algorithm. **The focus of our approach lies in how, for a model growth process based on operators and schedules to obtain a target model with given parameters, we can define and determine its optimal path without going through training on all possible paths (rather than the method for finding the optimal path itself)**, whereas the selection of the optimal schedule has been rarely studied in the area of model growth.
> > > > >
> > > > > In this paper, the schedule for the target model with 1.1B parameters consists of 168 paths. Assuming that it takes an average of 100 hours to train a target model through one schedule, if we aim to determine the schedule with the shortest train time and good model performance, we would need to train each schedule and then compare every target model to obtain the optimal schedule. This process would require $168 * 100 = 168,100 hours$, equivalent to **19.19** years. This is obviously unrealistic.
> > > > >
> > > > > Our method, on the other hand, allows us to find the optimal schedule for a target model without requiring training, as long as we follow the theory we propose. By utilizing SLOP, we can directly determine the optimal schedule and only need to train one target model(in paper’s case, 99 hours), which has a short training cost and good performance. Furthermore, we have validated the correctness of our theory through actual training comparable experiments, and we believe that this is the significance of the SLOP optimization.
> > > > >
> > > > > ---

---

> > > > > > ### Comment · Reviewer_ksYf · 2024-11-26
> > > > > > **Response by Reviewer ksYf**
> > > > > >
> > > > > > Thank you for your detailed response. As I understand it, the main contribution of the proposed method, SLOP, lies in two aspects:
> > > > > >
> > > > > > 1. Formulating the problem of identifying the optimal growth path and simplifying it to finding the total number of parameter changes through mathematical derivation.
> > > > > >
> > > > > > 2. Using an efficient DP or Dijkstra algorithm to solve the simplified problem.
> > > > > >
> > > > > > However, I remain concerned about the validity or necessity of both parts:
> > > > > >
> > > > > > The mathematical derivation in the first part appears to be incorrect. Specifically, you are optimizing an upper bound in Equation 5. **Maximizing an upper bound does not necessarily maximize the original objective, rendering the derivation unjustified**. Furthermore, as of the time of this response, the notation issues in the manuscript have not been addressed.
> > > > > >
> > > > > > Assuming, for the sake of argument, that the derivation in the first part is correct, your approach can approximate the training cost using the total number of parameter changes. At this stage, there are only a few thousand possible solutions, which can be enumerated directly without requiring GPU training.
> > > > > > My question remains: why use a Dijkstra algorithm instead of straightforward enumeration? While the Dijkstra algorithm might appear fancy and contribute to the method's novelty, the problem is computationally trivial, and using Dijkstra only adds complexity without tangible benefit. Furthermore, I am confused by the repeated claim that this process requires significant GPU hours. That's nonsense and unrelated to my original question.
> > > > > >
> > > > > > Despite back-and-forth communications with the authors, I feel the discussion does not progress effectively. I have repeatedly posed and clarified the same concerns, but the authors' responses have either been irrelevant or fundamentally incorrect.
> > > > > > Based on my current assessment, I maintain my recommendation for rejection.

---

### Official Review · Reviewer_AQBp · 2024-11-02

**Soundness:** 2
**Presentation:** 3
**Contribution:** 3
**Rating:** 6
**Confidence:** 4

**Summary:**

This paper presents SLOP, a methodology for determining optimal schedules for growing smaller pre-trained language models into larger ones through multi-stage expansion. The key contribution is formulating the schedule optimization as a dynamic programming problem that balances training costs and model performance. The authors show how marginal utility (basically ratio of performance to time spent training) can be used as an appropriate measure for finding optimal schedules theoretically, without requiring extensive experimental training. Specifically, starting from a smaller model, this technique, scales (in stages) the model to a larger size (by altering number of layers layers, multi-head attention , feed-forward network , and hidden states). This is validated by growing the model from 100M to 1B parameters in 5-stages.

The core idea can be visualized as a graph problem where each "node" represents a possible model configuration (with specific hidden dimensions, FFN dimensions, layers, heads), and the "edges" represent the growth operations to transition between configurations. The "weight" of each edge corresponds to the number of parameters added by that growth operation. The intuition here is that the last numbers of parameter change are proportional to the least compute required.

**Strengths:**

1. The technical approach has merit in its mathematical formulation, showing how schedule optimization can be reformulated as a dynamic programming problem.
2. Theoretical work for optimizing model growth schedules that moves beyond empirical approaches.
3. Well-motivated use of marginal utility as an optimization metric that effectively connects model performance with training costs.

**Weaknesses:**

1. Limited number of growth stages (5) constrains the practical applicability of the approach.
2. Evaluation focused primarily on one architecture (GPT-2) despite broader claims about transformer-based LLMs.
3. Choice of initialization size (100M parameters) may miss important dynamics that could be studied at smaller scales (e.g., starting from 10M parameters).
4. Lack of justification for downstream task selection - the paper would benefit from comparing its evaluation tasks with those used in related work (e.g., MSG and ELLE etc).
5. Unclear explanation of how Cost/Time relates to number of parameters in the marginal utility calculations.
6. Figure 1 needs significant improvement to better illustrate the growth process.
7. The hardcoding of head numbers may limit adaptability to different architectures.

**Questions:**

1. Why was the number of stages limited to 5? Could the approach be extended to handle more stages?
2. Would the results hold if experiments were conducted starting from smaller models (e.g., 10M parameters)?
3. How does the approach generalize to different transformer architectures beyond GPT-2?
4. Could you provide more details on how the Cost/Time relationship in MUS was determined?
5. What was the rationale for selecting the specific downstream tasks used in evaluation?

---

> ### Author Response · Authors · 2024-11-20
>
> Thank you very much for your valuable comments and questions. I appreciate the time and effort you have put into reviewing our manuscript. Below, I address your concerns and provide further clarifications.
>
> ---
>
> **Q1**: Why was the number of stages limited to 5? Could the approach be extended to handle more stages?
>
> **A1**: We appreciate the opportunity to provide clarification on this concern. The growth stages adhere to current research on model growth methods, which typically expend one dimension of Transformer each time. Since the prevailing architecture of existing LLMs is Transformer, which essentially has four potential dimensions for expansion (hidden_dim, multi-head number, ffn_dim, and layer), existing model growth methods typically consider expanding only one dimension at a time'[1, 2]', resulting in a maximum of five expansion stages. We are following the existing work and do not consider complex cases when multiple growth dimensions can combine at the same stage and execute more than once, which may have more than 5 stages. We will leave this for our future work.
>
> > _[1] Gesmundo, Andrea, and Kaitlin Maile. "Composable function-preserving expansions for transformer architectures." arXiv preprint arXiv:2308.06103 (2023)._
>
> > _[2] Yao, Yiqun, et al. "2x faster language model pre-training via masked structural growth." arXiv preprint arXiv:2305.02869 (2023)._
>
> ---
>
> **Q2**: Would the results hold if experiments were conducted starting from smaller models (e.g., 10M parameters)?
>
> **A2**: Thank you for your highly valuable suggestions. We have conducted supplementary experiments on smaller models (from 27M to 100M) to verify the generality of our method. The experimental results presented in the table demonstrate that our method is equally applicable to models with smaller parameter sizes.
>
> |     | Initial  |  Stage1   |  Stage2   |  Stage3   |   Sum   |
> | --------:   | :----:   | :----:  | :----:  | :----:  |:----:  |
> |Metrics | FLOPs(e18)/wall time(h) |FLOPs(e18)/wall time(h) |FLOPs(e18)/wall time(h) |FLOPs(e18)/wall time(h) |FLOPs/wall time(h) | |
> | **ELLE-100M**    | (384,1024,6) |  (512,1536,8)  | (640,1536,10) | (768,2048,12)    | |
> | **ELLE-100M**    | 0.51/0.35 |  0.85/0.59    | 1.28/0.89   |  1.82/1.27   | 4.46/3.1|
> | **GPT-100M**    | (768,2048,12) | (768,2048,12)|(768,2048,12)|(768,2048,12)| |
> | **GPT-100M**    | 1.66/1.15 |  1.66/1.15 | 1.66/1.15| 1.66/1.15 |6.64/4.6|
> | **SLOP-100M**    |(384,1024,6)|(768,1024,6)|(768,2048,6)|(768,2048,12)| |
> | **SLOP-100M**    | 0.46/0.32 |  0.97/0.68  |  0.99/0.68   |  1.66/1.15   |   **4.08/2.83**|
>
> ---
>
> **Q3**: How does the approach generalize to different transformer architectures beyond GPT-2?
>
> **A3**: Thank you for providing us this opportunity to elaborate. The latest research on model growth and scaling laws has mostly focused on generative large models (decoder-only and GPT-like), as the newly released powerful LLMs are decoder-only architectures. Consequently, our research centers on decoder-only models, aiming to address issues such as forgetting and inefficiency encountered during the practical training of LLMs.
>
> Our base model adopts the llama structure, leveraging its leading position in the field of LLMs.
> Llama introduces certain modifications to the GPT-2 structure, including pre-layer normalization, RMSNorm normalization function, SwiGLU activation function, and rotated positional embeddings. These changes do not affect the number of parameters and, consequently, do not impact the results of our method.
>
> Although we haven’t conducted extra experiments, theoretically, our approach is equally applicable to models with different transformer-based structures.

---

> > ### Comment · Reviewer_AQBp · 2024-11-26
> > **A2 clarification**
> >
> > Could you please clarify the Table in the A2? I am not sure what part of this table is for smaller models (ie, 27M).

---

> > > ### Author Response · Authors · 2024-11-26
> > >
> > > Thank you for your quick response. In Table 2, SLOP-100M and ELLE-100M employ different schedules and operators for model growth, progressively increasing from an initial stage of 27M parameters with dimensions (384, 1024, 6) to Stage 3 of 105M parameters with dimensions(768, 2048, 12). GPT-100M serves as a comparison baseline, with a fixed number of parameters at 100M and constant model size throughout in each stage.

---

> > > ### Author Response · Authors · 2024-12-03
> > >
> > > Since the rebuttal phase is coming to an end, could you please let us know if our responses and clarification address the remained issues? We would greatly appreciate any further suggestions or clarifications you may have and are happy to discuss them further if needed.
> > >
> > > Thank you again for your time and consideration.

---

> ### Author Response · Authors · 2024-11-20
>
> **Q4**: What was the rationale for selecting the specific downstream tasks used in evaluation? the paper would benefit from comparing its evaluation tasks with those used in related work (e.g., MSG and ELLE etc).
>
> **A4**: In our paper, the downstream tasks selected are based on the technical reports of current industry-leading LLMs such as llama`[1]`, qwen`[2]`, specifically targeting several general downstream tasks for evaluating the capabilities of LLMs.
>
> To further evaluate SLOP's effectiveness, we ran additional experiments comparing SLOP to the model growth baseline ELLE on numerous downstream tasks described in the ELLE paper. Due to time constraints, we conducted our experiments using the models with 100M parameters, and the findings are shown in the table below.
>
> |     |WB|WB|NEWS|NEWS|REV|REV|BIO|BIO|CS|CS|Avg.|
> | --------:   | :----:   | :----:  | :----:  | :----:  |:----:| :----: | :----:  | :----:  | :----:  |:----: |:----: |
> | |MNLI|QNLI|Hyper|Ag|Helpness|IMDB|CHEM|RCT|ACL-ARC|SCIERC||
> |**ELLE**|78.12|83.77|78.75|**93.21**|86.59|92.81|79.98|87.00|73.43|79.79|83.35|
> |**SLOP**|**79.60**|**84.34**|**81.68**|93.12|**87.16**|**93.57**|**81.27**|**87.40**|**78.13**|**82.08**|**84.84**|
>
> > _[1] Touvron, Hugo, et al. "Llama: Open and efficient foundation language models." arXiv preprint arXiv:2302.13971 (2023)._
>
> > _[2] Yang, An, et al. "Qwen2 technical report." arXiv preprint arXiv:2407.10671 (2024)._
>
> ---
>
> **Q5**: Figure 1 needs significant improvement to better illustrate the growth process.
>
> **A5**: Thank you for your sincere suggestions on the figures, which would greatly enhance the quality of our paper. We have attempted to redraw Figure 1, which is now uploaded in the supplementation(`growth_process_supplementary.pdf in Supplementary Material`). We hope it can better illustrate the growth process.
>
> ---
>
> **Q6**: The hardcoding of head numbers may limit adaptability to different architectures.
>
> **A6**: Since the attention head numbers do not lead to changes in parameters, it consequently has no impact on the training time. In practical operations, we can set the appropriate head number based on the actual requirements. We further investigate how varying the number of attention heads affects the target model’s performance and downstream experiments, as detailed in Appendix C.2.
>
> ---
>
> **Q7**: Could you provide more details on how the Cost/Time relationship in MUS was determined?
>
> **A7**: We borrow the concept of Marginal Utility in economics`[1]` and propose to use Marginal Utility of schedule (MUS) as the measurement. In a nutshell, marginal utility is employed in economics to balance benefits and costs. The higher the benefit and the lower the cost, the higher the benefit/cost ratio, showing that the system delivers more value. In our case, we use MUS to balance model performance (benefit) in reducing PPL against training time (cost), which is specified by the optimization objective.
>
> > _[1] Paul A. Samuelson. A Note on Measurement of Utility. The Review of Economic Studies, 4(2): 155–161, 02 1937. ISSN 0034-6527. doi: 10.2307/2967612. URL https://doi.org/10. 2307/2967612._

---

### Official Review · Reviewer_Zsri · 2024-11-02

**Soundness:** 3
**Presentation:** 3
**Contribution:** 3
**Rating:** 6
**Confidence:** 3

**Summary:**

This study introduces an approach to model growth schedules for transformer-based large language models (LLMs). Unlike existing work that primarily focuses on the growth operators, this approach explores multi-stage growth schedules where each stage systematically expands various dimensions of the model—layer count, multi-head attention, feed-forward network dimensionality, and hidden layer size. The proposed method, Schedule Learning via Optimal Path (SLOP), borrows the concept of marginal utility from economics to determine an optimal schedule that balances training costs and model performance after each growth stage. By applying this measure, the problem of finding the best growth path is framed as a dynamic programming task, which is efficiently solved in polynomial time using an optimal path algorithm. Empirical results demonstrate that SLOP enhances key performance metrics such as loss and perplexity while also reducing overall training time. This suggests that SLOP can lead to more cost-effective training processes without compromising and even improving model performance.

**Strengths:**

Originality: Unlike traditional approaches that focus on growth operators, this work takes a unique approach by studying growth schedules. By framing model growth as a pathfinding problem guided by the marginal utility of each stage, the study provides a method to expand model size with reduced perplexity and without increasing training costs. This approach offers a fresh perspective on optimizing model development, shifting the focus from how models grow to when and in what order they expand.

Quality: The technical quality is solid, with thorough development and clear analysis of the proposed methods. The empirical results support the authors' claims about the efficiency and effectiveness of their approach in optimizing growth schedules.

Clarity: The paper is well-organized and clearly written, with coherent explanations of the background, literature, methodology, experimental setup, and results. This structure enhances readability and helps convey the research contributions effectively.

Significance: This research is significant for its potential to reduce the computational burden of trial-and-error training in an exponentially large search space. By optimizing growth schedules, the study provides insights that could make model training more cost-effective and accessible, which is particularly impactful for scaling large language models.

**Weaknesses:**

1) On the Choice of Target Structure in Table 2
It’s unclear why the authors chose only one target structure (2816, 7680, 8) for evaluation. This raises the question of whether the proposed method can be generalized to other target structures with different dimensions. It would be helpful for the authors to either justify this choice or provide additional experiments demonstrating the method's adaptability to a variety of target structures. This would help show that the approach is not limited to a specific configuration and can be applied more broadly.

2) Inflexible Target Structure
The current approach relies on a predefined target structure. Instead, could it be possible to allow the model to grow flexibly within a given duration 𝑇 and without a fixed target structure? This would enable the model to expand within computational budgets while still achieving satisfactory performance.

Minor Comments:
1) Font Size in Figures
The font size in almost all figures is too small, making it difficult for readers to follow the visual data and conclusions. Increasing the font size, especially in key charts and illustrations, would improve readability and accessibility, allowing readers to better understand and interpret the results presented.

**Questions:**

1) Correlation Between Training Times in Figure 3
The relationships between training times across different schedules in Figure 3 seem unclear, making it challenging to interpret. Providing a more detailed description and analysis would help readers understand how training time varies across different growth schedules and how it correlates with performance. This additional analysis could include specific comparisons or visual indicators to make the trends easier to follow.

2) Possibility of Finer-Granularity Stages
A question remains on whether the current approach supports finer-grained growth stages, such as incrementally increasing the layer count at each stage. Exploring this would add flexibility to the model growth process, potentially allowing smoother transitions and more granular control over resource allocation at each stage. Clarifying whether the method could accommodate such finer stages would help readers understand its adaptability to different training strategies on model growth schedules.

3) Details on Measuring GPU Wall Time
It is unclear how GPU wall time was measured across different stages. Specifically, what are the defined start and end times for each stage? Providing this information will clarify how the measurement was conducted and ensure the results can be reproduced accurately.

---

> ### Author Response · Authors · 2024-11-22
>
> Thank you for your elaborate reviews and suggestions. We summarize your questions and reply to them as follows, and we are happy to address any further feedback!
>
> ---
>
> **Q1**: On the Choice of Target Structure in Table 2 It’s unclear why the authors chose only one target structure (2816, 7680, 8) for evaluation. This raises the question of whether the proposed method can be generalized to other target structures with different dimensions.
>
> **A1**: Due to constraints in time and computational power, Table 2 only presents a comparison with baselines in only one target structure. To demonstrate the applicability of our method to all target structures, we have supplemented experiments with an additional target structure(2048, 5632, 16), as shown in the following table. The experiments further corroborate the versatility of SLOP across various target structures.
>
> |     | PPL  | Time(GPU hours) |
> | --------:   | :----:   | :----:  |
> |SCHL-single stage |32 |172|
> |SCHL-MSG|36|119|
> |ELLE|34|114|
> |SLOP|34|108|
>
> ---
>
> **Q2**: The current approach relies on a predefined target structure. Instead, could it be possible to allow the model to grow flexibly within a given duration 𝑇 and without a fixed target structure?
>
> **A2**: Firstly, the proposed approach does not rely on one predefined target structure. As illustrated in Figure 1.(c), there could be more than two target structures for the same parameter LLM: those of (2816,7680,8),(1536,4096,32),(1280, 3584, 10, 40), (1792, 4864, 14, 20) and so on.
>
> Second, we believe it is theoretically feasible to allow the model to grow flexibly within a certain druation T without a fixed target structure. However, according to published technical reports, such as llama`[1]`, qwen`[2]`, baichuan`[3]`, and mistral`[4]`, existing LLMs often comply with specific constraints, which may be mainly due to the GPU parallel strategies. These constraints include:
> 1. The hidden dimension size is a multiple of 128.
> 2. The hidden dimension is either 8/3 or 4 times the ffn dimension.
> 3. The number of attention heads should be divisible by the hidden dimension; nevertheless, this has no effect on the model’s size.
>
> Therefore, in our current experimental setup, we strictly adhere to these constraints and have not taken into account all possible scenarios. Furthermore, as mentioned in the Limitation section, there could be more complex cases where multiple growth dimensions can combine at the same stage and execute more than once. We leave this for future work.
>
> > _[1] Touvron, Hugo, et al. "Llama: Open and efficient foundation language models." arXiv preprint arXiv:2302.13971 (2023)._
>
> > _[2] Yang, An, et al. "Qwen2 technical report." arXiv preprint arXiv:2407.10671 (2024)._
>
> > _[3] Yang, Aiyuan, et al. "Baichuan 2: Open large-scale language models." arXiv preprint arXiv:2309.10305 (2023)._
>
> > _[4] Jiang, Albert Q., et al. "Mistral 7B." arXiv preprint arXiv:2310.06825 (2023)._
>
> ---
>
> **Q3**: Font Size in Figures The font size in almost all figures is too small, making it difficult for readers to follow the visual data and conclusions. Increasing the font size, especially in key charts and illustrations, would improve readability and accessibility.
>
> **A3**: Thank you for your sincere suggestions on the font size in the figures, which would help a lot for the quality of our work. We will increase the font size to ensure that readers can easily understand the visual data and conclusions presented in the revised version.
>
> ---

---

> ### Author Response · Authors · 2024-11-22
>
> **Q4**: Correlation Between Training Times in Figure 3 The relationships between training times across different schedules in Figure 3 seem unclear, making it challenging to interpret.
>
> **A4**: Thanks for giving us the chance to clarify this concern. In Figure 3, we have calculated the correlation among the lists of 4-stage training times of models with different $\prod \Delta params$ (expanded parameters through different schedules). A higher correlation value (indicated by a color closer to red) suggests that the computational costs among these models are more similar. As observed in the figure, if the values of $\prod \Delta params$ between two models are closer, the corresponding color they map to in the correlation heatmap tends to be closer to red, particularly noting the prominent diagonal trend dominated by red color. These visual cues align with our theoretical expectations.
>
> ---
>
> **Q5**: Possibility of Finer-Granularity Stages A question remains on whether the current approach supports finer-grained growth stages, such as incrementally increasing the layer count at each stage.
>
> **A5**: This is a great point. In accordance with commonly referred to model growth methods`[1, 2]`, we limit SLOP to just expanding one dimension at each stage.
>
> It would be interesting to explore the more complex cases, including finer-grained growth stages and other cases, as we have mentioned in the Limitations in Appendix A.However, we believe the situations should also adhere to the constraints outlined in our response to your Question 2. We'll leave this for future work.
>
> > _[1] Gesmundo, Andrea, and Kaitlin Maile. "Composable function-preserving expansions for transformer architectures." arXiv preprint arXiv:2308.06103 (2023)._
>
> > _[2] Yao, Yiqun, et al. "2x faster language model pre-training via masked structural growth." arXiv preprint arXiv:2305.02869 (2023)._
>
> ---
>
> **Q6**: Details on Measuring GPU Wall Time It is unclear how GPU wall time was measured across different stages. Specifically, what are the defined start and end times for each stage? Providing this information will clarify how the measurement was conducted and ensure the results can be reproduced accurately.
>
> **A6**: In our paper, the GPU Wall Time refers to the GPU time necessary to complete the model training in each stage. The calculation formula for each stage is:
>
> $$GPU\\_Wall\\_Time = \frac{The\\_total\\_number\\_of\\_floating - point\\_operations}{The\\_number\\_of\\_GPUs \times GPU\\_peak\\_FLOPs \times GPU\\_utilization }$$
>
> The total number of floating-point operations is calculated using`[1]`, and the number and utilization rates of GPUs are based on the actual value during training.
>
> > _[1] Narayanan, Deepak et al. "Efficient Large -Scale Language Model Training on GPU Clusters Using Megatron-LM",Proceedings of the International Conference for High Performance Computing, Networking, Storage and Analysis abs/2104.04473 (2021): 1-15._
>
> ---

---

> ### Author Response · Authors · 2024-11-26
>
> We sincerely thank you very much for these constructive comments and evaluation of our manuscript. As the discussion phase end time has been postponed, we would like to kindly ask you to take a look at our responses and reevaluate our work based on our clarifications. Please let us know whether our response addresses your concerns or whether there is any further detail we can provide to help address these concerns.
>
> Thank you again for dedicating your time to reviewing our paper.

---

> > ### Comment · Reviewer_Zsri · 2024-11-27
> >
> > Thank you, authors, for addressing my questions and concerns. I appreciate the clarity provided and find the explanations satisfactory. I will maintain my positive scores.

---

> > > ### Author Response · Authors · 2024-11-27
> > >
> > > Thanks for your positive feedback. We will carefully address your concerns in our new version.

---

### Official Review · Reviewer_Fq5M · 2024-11-05

**Soundness:** 3
**Presentation:** 3
**Contribution:** 3
**Rating:** 8
**Confidence:** 3

**Summary:**

The authors propose a method to incrementally grow a larger model from a smaller model. The authors do so by measuring marginal utility at each stage. They test their method on three LLMs on a well known benchmark.

**Strengths:**

- It is a mature work that seems to be mathematically derived. The proofs are mature and solid and the results are good.
- putting structure on model growth is an art rather than a science, and the authors have done a good job at trying to propose a good local optimization.

**Weaknesses:**

- missing more than one LLM in experiments
- missing code, limitations, future work sections

**Questions:**

- Model growth as a way to lessen the burden of training compute / time. Could be very significant as far as pretraining is concerned.
- “At each stage, one dimension is expanded to develop an intermediate structure until the
entire target LLM structure is attained.” – is dimension really the right term for the growth target?
- I’m not sure why change in t ⇔ change in params
- Does this bias your algo towards operators that incur the lowest growth in params?
- Have you considered picking a math symbol for “params”? (It’s not theta, is it?)
- I understand the broad strokes of your proof, but there is enough difficulty in notation and skipped steps that its hard to agree with it outright. Perhaps more explanation or reminders of the terms would be helpful.
- By your pseudocode, this algo appears greedy to an extent (always choosing the vertex satisfying the minimum distance.) Can you comment on this? Have you considered inserting noise?
- There are some dimensions that you haven’t considered (whether to train in sparsity to layers, modularity in layers wrt attention type, and perhaps some parameter quantization dimension), does this technique extend to them?
- I’m not sure what the starting model was and/or architectural decisions were. Are you borrowing an uninitialized Llama structure? Or are you starting so much from scratch that you’re just starting at a basic transformer?
- You have not included code, as far as I can see
- Results look good. What are the limitations of your work? Future steps?
- Overall readability is not so good. I would recommend at least passing this through ChatGPT!

---

> ### Author Response · Authors · 2024-11-20
>
> Thank you for the detailed and insightful discussions on our paper. We hope the following clarifications could provide more clear support for our claims and help address your concerns.
>
> ---
>
> **Q1**: At each stage, one dimension is expanded to develop an intermediate structure until the entire target LLM structure is attained.” – is dimension really the right term for the growth target?
>
> **A1**: Thanks for your insightful inquiry. We follow previous model growth works`[1,2]`, utilizing dimension to represent the model region where the growth operator operates. Furthermore, as mentioned in the Limitation section, we do not consider complex cases where multiple growth dimensions can combine at the same stage and execute more than once. We leave this for future work.
>
> > _[1] Gesmundo, Andrea, and Kaitlin Maile. "Composable function-preserving expansions for transformer architectures." arXiv preprint arXiv:2308.06103 (2023)._
>
> > _[2] Yao, Yiqun, et al. "2x faster language model pre-training via masked structural growth." arXiv preprint arXiv:2305.02869 (2023)._
>
> ---
>
> **Q2**: I’m not sure why change in t ⇔ change in params
>
> **A2**: Thank you for bringing up this important clarification point. We omit some of the reasoning steps for brevity in the paper. Given a fixed computing budget, there exists a positive correlation between the GPU time required for training and the FLOPs. Therefore, it can be stated that: $\Delta t = f(\Delta FLOPs)$. Based on the theory of scaling laws`[1]`: $FLOPs \approx 6ND$, where $N$ represents model size(params) and $D$ denotes the number of training tokens. When the training dataset remains unchanged, $D$ is a constant value. Therefore, $\Delta t = f(\Delta FLOPs) \approx g(\Delta params)$, while $t$ and $params$ exhibit the same trend of increase, we can conclude that:
>
> $$ \mathop{argmax}\limits_{\phi_k \in \overline{\epsilon}}{\sum_{k=1}^4 \frac{\Delta ppl_{\phi_k}}{\Delta t(\phi_k)}} \Longleftrightarrow \mathop{argmax}\limits_{\phi_k \in \overline{\epsilon}}{\sum_{k=1}^4 \frac{\Delta ppl_{\phi_k}}{\Delta params(\phi_k)}}$$
>
> > _[1] Kaplan, Jared, et al. "Scaling laws for neural language models." arXiv preprint arXiv:2001.08361 (2020)._
>
> ---
>
> **Q3**: Does this bias your algo towards operators that incur the lowest growth in params?
>
> **A3**: We don't quite understand this question. What does bias refer to? We hope the following clarifications will help answer this question.Our algorithm explores schedules obtained by considering different growth operators' orderings with the aim of finding the product of minimal parameter variations($\prod \Delta params$). We would be happy to address this further with the reviewer.
>
> ---
>
> **Q4**: Have you considered picking a math symbol for “params”? (It’s not theta, is it?)
>
> **A4**: Thank you for your suggestion. To make the work more readable, we will consider using N to denote the parameters (params). Yes, it's not theta.
>
> ---
>
> **Q5**: There is enough difficulty in notation and skipped steps that its hard to agree with it outright. Perhaps more explanation or reminders of the terms would be helpful.
>
> **A5**: Thank you for your suggestion. In the upcoming version, we will incorporate the methodology along with additional descriptions to enhance the clarity of the presented arguments.
>
> ---
>
> **Q6**: By your pseudocode, this algo appears greedy to an extent (always choosing the vertex satisfying the minimum distance.) Can you comment on this? Have you considered inserting noise?
>
> **A6**: We have selected a commonly used algorithm for finding the optimal path, namely the Viterbi algorithm. Of course, other algorithms that identify optimal paths are equally applicable. Could you please clarify what you mean by "inserting noise"? We would like to discuss this further.
>
> ---
>
> **Q7**: There are dimensions that haven’t considered (whether to train in sparsity to layers, modularity in layers wrt attention type, and perhaps some parameter quantization dimension), does this technique extend to them?
>
> **A7**: Existing model growth methods generally consider the following four dimensions for expansion: hidden_dim, head_num, ffn_dim, and layer, and only one dimension is chosen for expansion each time`[1,2]`. Therefore, our research aims to explore methods for obtaining optimal schedules within the constraints of current research on model growth operators. There are other expanding dimensions such as key/query/value dimensions, however, these dimensions typically adopt fixed values based on practical experience (e.g., head dimension = hidden_dim / head_num) and do not significantly impact the number of parameters, hence they are not within the scope of our consideration.
>
> > _[1]Gesmundo, Andrea, and Kaitlin Maile. "Composable function-preserving expansions for transformer architectures." arXiv preprint arXiv:2308.06103 (2023)._
>
> > _[2]Yao, Yiqun, et al. "2x faster language model pre-training via masked structural growth." arXiv preprint arXiv:2305.02869 (2023)._

---

> > ### Author Response · Authors · 2024-11-20
> >
> > **Q8**:  I’m not sure what the starting model was and/or architectural decisions were. Are you borrowing an uninitialized Llama structure? Or are you starting so much from scratch that you’re just starting at a basic transformer?
> >
> > **A8**:  Our base model adopts the llama architecture and training from scratch using 25B tokens, and upon this foundation, we proceed with subsequent model growth training.
> >
> > ---
> >
> > **Q9**:  You have not included code, as far as I can see
> >
> >
> > **A9**:  The code is provided in the Supplementary Material and can be accessed directly from the review page.
> >
> > ---
> >
> > **Q10**:  Results look good. What are the limitations of your work? Future steps?
> >
> > **A10**:Thank you for your insightful inquiry. As briefly stated in Section 5 Conclusion and Appendix A Limitations, our work includes the following limitations:
> > 1. we do not consider complex cases when multiple growth dimensions can combine at the same stage and execute more than once.
> > 2. While not affecting the overall conclusions, there exist deviations between the experimental results and our inferences. For instance, among different target models, models with larger $\prod \Delta params$  require less training time.
> > 3. Due to limited computing capacity and budget, the largest models in our experiments have 1 billion parameters, which is a significant difference from existing LLMs. This constraint is present in the vast majority of research projects, according to our knowledge.
> >
> > These are the issues that require further research and analysis in our future work.
> >
> > ---
> >
> > **Q11**: Overall readability is not so good. I would recommend at least passing this through ChatGPT!
> >
> > **A11**: Thank you for pointing this out.  We will use ChatGPT or other tools to enhance the clarity and readability of our subsequent version.

---

> ### Author Response · Authors · 2024-11-26
>
> We sincerely thank you very much for these constructive comments and evaluation of our manuscript. As the discussion phase end time has been postponed, we would like to kindly ask you to take a look at our responses and reevaluate our work based on our clarifications. Please let us know whether our response addresses your concerns or whether there is any further detail we can provide to help address these concerns.
>
> Thank you again for dedicating your time to reviewing our paper.

---

> > ### Comment · Reviewer_Fq5M · 2024-12-03
> >
> > Hey team, thanks for looking over my questions. Admittedly, some were ill posed. I'll increase the score.

---

> ### Author Response · Authors · 2024-12-03
>
> Thank you for your thoughtful and positive feedback on our rebuttal. We are truly grateful for your valuable suggestions, and will carefully follow your advice, incorporating these discussions into the final version of the paper.
>
> Once again, we sincerely appreciate your constructive comments and support throughout the review process.

---

### Author Response · Authors · 2024-11-27
**Response for general Concern**

## Revised Paper

In general, we express our gratitude to the reviewers for their invaluable feedback, and have revised and re-uploaded the paper based on the reviewers' suggestions. The main changes are noted in yellow. The updated part primarily includes:
+ Clarify the relaxation method and notation we used in the Methodology section.
+ Adding experiment of exploring the impact of different target model structures in Appendix C.3.
+ Adding experiment of employing SLOP on smaller parameters. in Appendix C.4.
+ Adding experiment of performance on the downstream tasks compared with baselines in Appendix D.

---

### Note · Authors · 2025-01-23

I have read and agree with the venue's withdrawal policy on behalf of myself and my co-authors.